# Analyses of gut microbiota and plasma bile acids enable stratification of patients for antidiabetic treatment

Yanyun Gu [1], Xiaokai Wang[2,3], Junhua Li [2,4], Yifei Zhang[1], Huanzi Zhong [2,5], Ruixin Liu[1], Dongya Zhang[2], Qiang Feng[2], Xiaoyan Xie[1], Jie Hong[1], Huahui Ren[2,4,6], Wei Liu[7], Jing Ma[7], Qing Su[8], Hongmei Zhang[8], Jialin Yang[9], Xiaoling Wang[10], Xinjie Zhao[10], Weiqiong Gu[1], Yufang Bi[1], Yongde Peng[11], Xiaoqiang Xu[2,3], Huihua Xia[2,4], Fang Li[2,4,6], Xun Xu[2], Huanming Yang[2,12], Guowang Xu [10], Lise Madsen [2,3,13], Karsten Kristiansen [2,5], Guang Ning[1] & Weiqing Wang[1]

Antidiabetic medication may modulate the gut microbiota and thereby alter plasma and faecal bile acid (BA) composition, which may improve metabolic health. Here we show that treatment with Acarbose, but not Glipizide, increases the ratio between primary BAs and secondary BAs and plasma levels of unconjugated BAs in treatment-naive type 2 diabetes (T2D) patients, which may beneficially affect metabolism. Acarbose increases the relative abundances of *Lactobacillus* and *Bifidobacterium* in the gut microbiota and depletes *Bacteroides*, thereby changing the relative abundance of microbial genes involved in BA metabolism. Treatment outcomes of Acarbose are dependent on gut microbiota compositions prior to treatment. Compared to patients with a gut microbiota dominated by *Prevotella*, those with a high abundance of *Bacteroides* exhibit more changes in plasma BAs and greater improvement in metabolic parameters after Acarbose treatment. Our work highlights the potential for stratification of T2D patients based on their gut microbiota prior to treatment.

[1] Shanghai National Research Centre for Endocrine and Metabolic Diseases, State Key Laboratory of Medical Genomics, Shanghai Institute for Endocrine and Metabolic Diseases, Ruijin Hospital, Shanghai Jiaotong University School of Medicine, 200025 Shanghai, China. [2] BGI-Shenzhen, China National GeneBank-Shenzhen, 518083 Shenzhen, China. [3] BGI Education Centre, University of Chinese Academy of Sciences, 518083 Shenzhen, China. [4] Shenzhen Key Laboratory of Human commensal microorganisms and Health Research, BGI-Shenzhen, Shenzhen 518083, China. [5] Laboratory of Genomics and Molecular Biomedicine, Department of Biology, University of Copenhagen, 2100 Copenhagen, Denmark. [6] Shenzhen Engineering Laboratory of Detection and Intervention of human intestinal microbiome, BGI-Shenzhen, Shenzhen 518083, China. [7] Renji Hospital affiliated to Shanghai Jiaotong University Medical School, 200127 Shanghai, China. [8] Xinhua Hospital affiliated to Shanghai Jiaotong University Medical School, 200092 Shanghai, China. [9] MinHang Central Hospital affiliated to Fudan University Medical School, 201100 Shanghai, China. [10] Dalian Institute of Chemical Physics, Chinese Academy of Science, 116011 Dalian, China. [11] Shanghai General Hospital, Shanghai Jiaotong University, 200080 Shanghai, China. [12] James D. Watson Institute of Genome Sciences, Hangzhou 310008, China. [13] National Institute of Nutrition and Seafood Research (NIFES), 5817 Bergen, Norway. Yanyun Gu, Xiaokai Wang, Junhua Li, Yifei Zhang, Huanzi Zhong and Ruixin Liu contributed equally to this work. Correspondence and requests for materials should be addressed to K.K. (email: kk@bio.ku.dk) or to G.N. (email: gning@sibs.ac.cn) or to W.W. (email: wqingw61@163.com)

The importance of the intestinal microbiota in host health and pathogenesis of several non-communicable diseases such as cancer, obesity and type 2 diabetes (T2D), and even mental disorders is now recognised[1, 2]. Cross-sectional case-control studies have revealed microbial dysbiosis in T2D patients[3, 4], possibly contributing to disease development. Although solid proof of a causal link between gut microbial dysbiosis and T2D is still missing, it has been reported that therapeutic strategies such as bariatric surgery[5, 6] and Metformin treatment[7–9] may confer metabolic benefits including improved glycaemic control through alteration of the gut microbiota. Furthermore, recent studies have demonstrated that single bacterial species associated with insulin resistance and obesity upon transfer to normal specific pathogen-free (SPF) mice confer the same phenotype to the recipient mice[10, 11].

Host-gut microbiota interaction is central in bile acid (BA) metabolism and signalling, and it is essential for the maintenance of metabolic health[12]. Additionally, compared to healthy individuals, alterations in the levels and compositions of plasma BAs have been observed in patients with diabetes or obesity[13–15], further highlighting a possible involvement of BA metabolism in the pathogenesis of metabolic diseases. Finally, an animal study suggests that BA signalling is required for mediating the therapeutic effects of bariatric surgery[16].

Acarbose, a pseudo-tetrasaccharide produced by *Actinoplanes* species and an alpha-glycosidase inhibitor, is recommended in China and other Asian countries as an alternative to Metformin for first line treatment of T2D[17–19]. Unlike classic oral antidiabetic drugs (OADs) such as thiazolidinediones or sulfonylureas (SUs)[20] with well-characterised molecular targets affecting insulin resistance or secretion, Acarbose exerts its hypoglycaemic effects by inhibiting the hydrolysis of complex carbohydrates on the brush boarder of the upper intestinal epithelium thereby reducing glucose absorption in this part of the intestine. This further leads to an increased concentration of complex carbohydrates in the lower part of the intestine[21], potentially altering microbial fermentation in the distal gut.

To elucidate how OADs may affect the gut microbiota, and discriminate such changes from disease-dependent alterations, intervention studies are needed on treatment-naive T2D patients admitted to antidiabetic treatment. In a multicentre, randomised and positively controlled clinical trial, where treatment-naive T2D patients were assigned to treatment with one of two drugs commonly used in China, Acarbose or the sulfonylurea Glipizide, to achieve targeted glycaemic control, we compared the effects of the two drugs on metabolic parameters, including plasma BAs, and the intestinal microbiota. We identify drug-dependent alterations in the microbiome and plasma BAs that correlate with clinical measurements and outcomes, including body mass index (BMI), blood glucose, glycated haemoglobin (HbA1c), lipid profiles, insulin secretion and resistance status. Of interest, treatment-naive T2D patients can be stratified into two groups based on their individual baseline microbiomes. Patients harbouring a gut microbiota with a higher baseline abundance of *Bacteroides* have lower levels of secondary BAs and exhibit more beneficial therapeutic responses to Acarbose treatment, including reduced BMI, and improved insulin resistance status and lipid profile, suggesting that baseline metagenome signatures may be used to stratify T2D patients prior to treatment.

## Results

**Baseline characteristics of patients and clinical outcomes.** To characterise the clinical effects of Acarbose and Glipizide, we

**Table 1 Clinical parameters in the pre-treatment and post-treatment samples in the two treatment arms**

| | Pre-treatment | | | Post-treatment | | | P value (Pre vs. Post) | |
| --- | --- | --- | --- | --- | --- | --- | --- | --- |
| | Acarbose (n = 51) | Glipizide (n = 43) | P value | Acarbose (n = 51) | Glipizide (n = 43) | P value | Acarbose | Glipizide |
| *Demographic characteristics* | | | | | | | | |
| Age (years) | 53 ± 7 | 54 ± 7 | 0.497 | | | NA | NA | NA |
| Gender (male/total) | 34/51 | 24/43 | 0.387 | | | NA | NA | NA |
| Body weight (kg) | 74.7 ± 10.3 | 71. 5 ± 10.7 | 0.067 | 72.1 ± 10.1 | 70.6 ± 10.2 | 0.190 | 2.36E−07 | 0.138 |
| BMI (kg/m²) | 26.3 ± 3.2 | 26.0 ± 3.4 | 0.265 | 25.4 ± 3.2 | 25.7 ± 3.3 | 0.945 | 3.48E−07 | 0.113 |
| WC (cm) | 91.5 ± 8.5 | 90.7 ± 8.4 | 0.527 | 88.6 ± 7.3 | 89.5 ± 8.4 | 0.870 | 3.84E−04 | 0.144 |
| WHR | 0.92 ± 0.06 | 0.92 ± 0.05 | 0.485 | 0.90 ± 0.05 | 0.90 ± 0.06 | 0.779 | 0.005 | 0.019 |
| *Glycaemic characteristics* | | | | | | | | |
| HbA1c (%) | 7.5 ± 0.8 | 7.7 ± 0.9 | 0.541 | 6.4 ± 0.5 | 6.3 ± 0.7 | 0.508 | 1.45E−09 | 1.82E−08 |
| FPG (mmol/L) | 7.6 ± 1.4 | 7.9 ± 1.4 | 0.291 | 6.6 ± 0.9 | 6.7 ± 1.2 | 0.665 | 3.58E−07 | 2.48E−06 |
| PPG (mmol/L) | 14.3 ± 2.7 | 15.0 ± 2.8 | 0.164 | 9.2 ± 1.8 | 10.8 ± 2.8 | 0.002 | 5.30E−10 | 6.29E−10 |
| AUC blood glucose (mmol/L*min) | 2241 ± 363 | 2381 ± 351 | 0.077 | 1507 ± 263 | 1804 ± 359 | 6.55E−05 | 5.30E−10 | 6.98E−11 |
| AUC Insulin (pmol/L*min) | 5981 ± 3340 | 5460 ± 2507 | 0.697 | 3700 ± 1989 | 6618 ± 3536 | 1.62E−06 | 2.40E−06 | 0.019 |
| AUC C Peptide (ng/mL*min) | 528 ± 210 | 501 ± 154 | 0.978 | 389 ± 143 | 575 ± 161 | 1.94E−07 | 1.94E−08 | 0.002 |
| HOMA-IR (μIU *mmol) | 3.7 ± 2.3 | 3.6 ± 2.1 | 0.868 | 2.6 ± 1.8 | 2.9 ± 2.1 | 0.638 | 0.002 | 0.050 |
| *Other clinical indices* | | | | | | | | |
| SBP (mmHg) | 127 ± 16 | 131 ± 18 | 0.431 | 121 ± 13 | 131 ± 19 | 0.128 | 0.022 | 0.043 |
| DBP (mmHg) | 82 ± 8 | 81 ± 10 | 0.675 | 75 ± 8 | 76 ± 9 | 0.703 | 2.94E−05 | 0.013 |
| Triglycerides (mmol/L) | 2.5 ± 2.1 | 2.1 ± 1.2 | 0.459 | 1.6 ± 1.2 | 1.7 ± 0.8 | 0.093 | 2.39E−05 | 0.382 |
| Total cholesterol (mmol/L) | 5.1 ± 1.2 | 4.9 ± 1.0 | 0.696 | 4.8 ± 1.2 | 4.8 ± 0.9 | 0.712 | 0.007 | 0.228 |
| LDL cholesterol (mmol/L) | 1.2 ± 0.4 | 1.2 ± 0.4 | 0.500 | 1.2 ± 0.3 | 1.3 ± 0.6 | 0.864 | 0.276 | 0.5355 |
| HDL cholesterol (mmol/L) | 3.1 ± 1.0 | 3.1 ± 0.8 | 0.492 | 3.0 ± 1.0 | 2.9 ± 0.8 | 0.514 | 0.484 | 0.0357 |
| FLI | 334 ± 554 | 237 ± 341 | 0.352 | 101 ± 147 | 1621 ± 271 | 0.400 | 1.92E−09 | 0.013 |

WC, waist circumference; WHR, waist-hip circumference ratio; HbA1c, glycated haemoglobin; FPG, fasting plasma glucose; PPG, postprandial plasma glucose; AUC, area under the curve; HOMA-IR, homeostasis model assessment of insulin resistance; LDL, low-density lipoprotein; HDL, high-density lipoprotein; FLI, fatty liver index: FLI = (e$^{0.953 \times \log_e [TG] + 0.139 \times BMI + 0.718 \times \log_e[\gamma GT]}$ + 0.053 × WC −15.745)/(1 + e$^{0.953 \times \log_e [TG] + 0.139 \times BMI + 0.718 \times \log_e [\gamma GT]}$ + 0.053 × WC −15.745) × 100. γGT, gamma-glutamyltransferase
Data are presented as mean ± SD

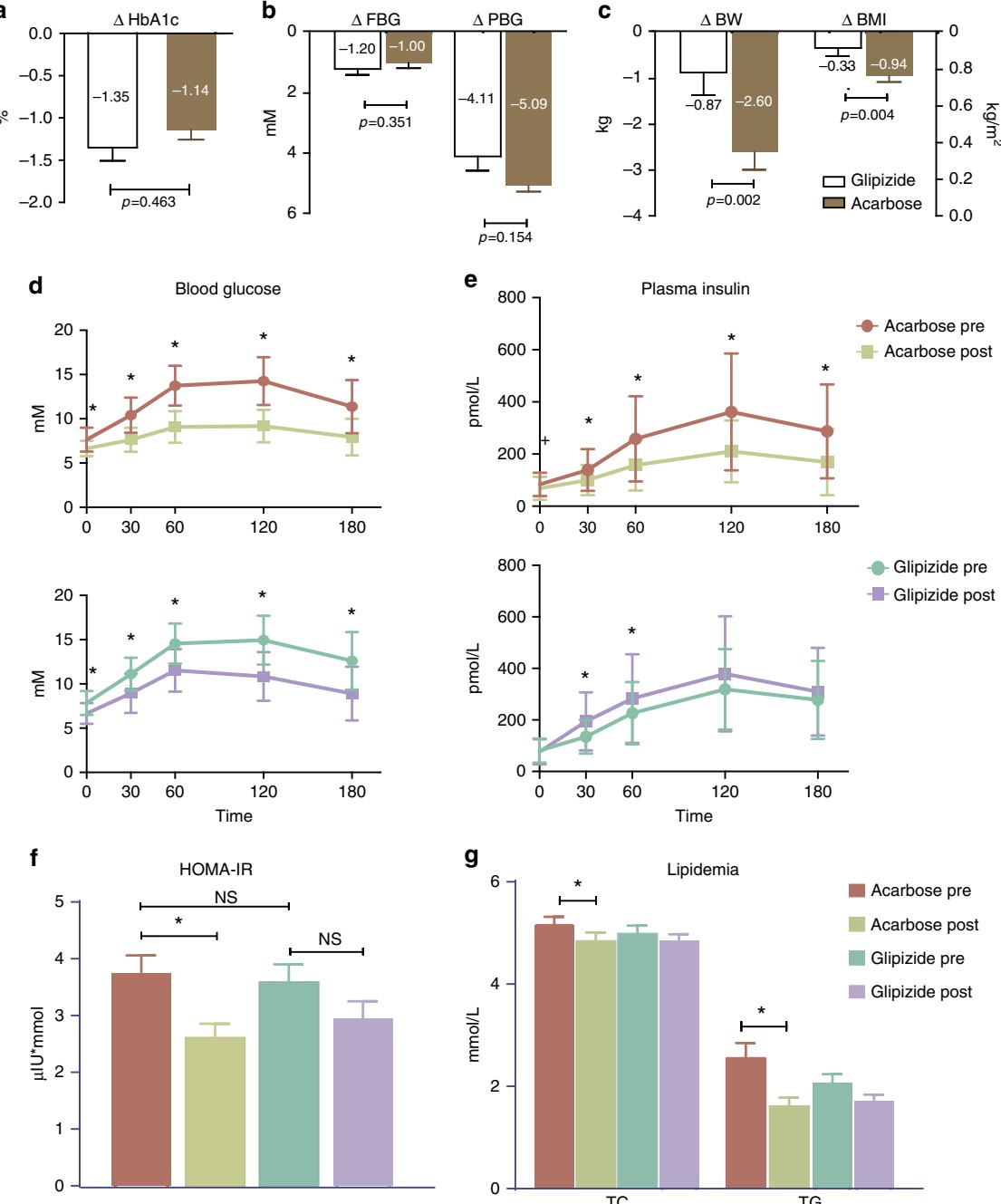

**Fig. 1** Major clinical outcomes in patients after 3 months of treatment with Acarbose or Glipizide. **a** Effect on glycated haemoglobin A1c (HbA1c), in the Acarbose (brown) and Glipizide (white) arms, Wilcoxon rank-sum test. **b** Effects on fasting blood glucose concentration (FBG), postprandial blood glucose concentration (PBG) in the Acarbose (brown) and Glipizide (white) treatment arms; paired Wilcoxon rank-sum test, *P < 0.01. **c** Effects on body weight (BW) and body mass index (BMI) in the Acarbose (brown) and Glipizide (white) treatment arms. **d** Blood glucose excursion curve and **e** plasma insulin release curve during the meal tests in the two treatment arms, paired Wilcoxon rank-sum test, *P < 0.01, + P < 0.05, pre-treatment vs. post-treatment with Acarbose. **f** Effects on HOMA-IR, **g** Total cholesterol (TC) and triglycerides (TG) in the two treatment arms; paired Wilcoxon rank-sum test, *P < 0.01. n = 51 in Acarbose pre and Acarbose post-treatment, n = 43 in Glipizide pre and Glipizide post-treatment

recruited patients from five centres in Shanghai, China. A total of 171 treatment-naive T2D patients were screened and based on inclusion and exclusion criteria (Supplementary Table 1; registered in ClinicalTrials.gov. NCT01758471), 106 patients were enrolled and 1:1 randomised into the two treatment arms (Supplementary Methods). There was no difference between the two arms in relation to gender, age and BMI (Table 1). Ninety-four of the enrolled patients completed the study and were included in the analyses, while 5 in the Acarbose group and 7 in the Glipizide

group were not included in the analysis (flow chart shown in Supplementary Fig. 1). The goal of the treatment was to achieve a generally recommended targeted level of HbA1c below 7.0%[22, 23] in both arms within 3 months, indicative of improved glycaemic control. Faecal and plasma samples were collected at baseline and after 3 months of treatment. There were no significant differences between anthropomorphic, metabolic, biochemical parameters and HbA1c levels in the two groups at baseline (Wilcoxon rank-sum test, P > 0.05, Table 1).

At the end of the 3-month treatment, the reductions in HbA1c levels, and fasting blood glucose (FBG) and postprandial blood glucose (PBG) levels were significant in both groups (paired Wilcoxon rank-sum test, $P < 0.01$ between baseline and treatment, Table 1), and both arms reached the targeted level of glycaemic control that did not differ significantly (Wilcoxon

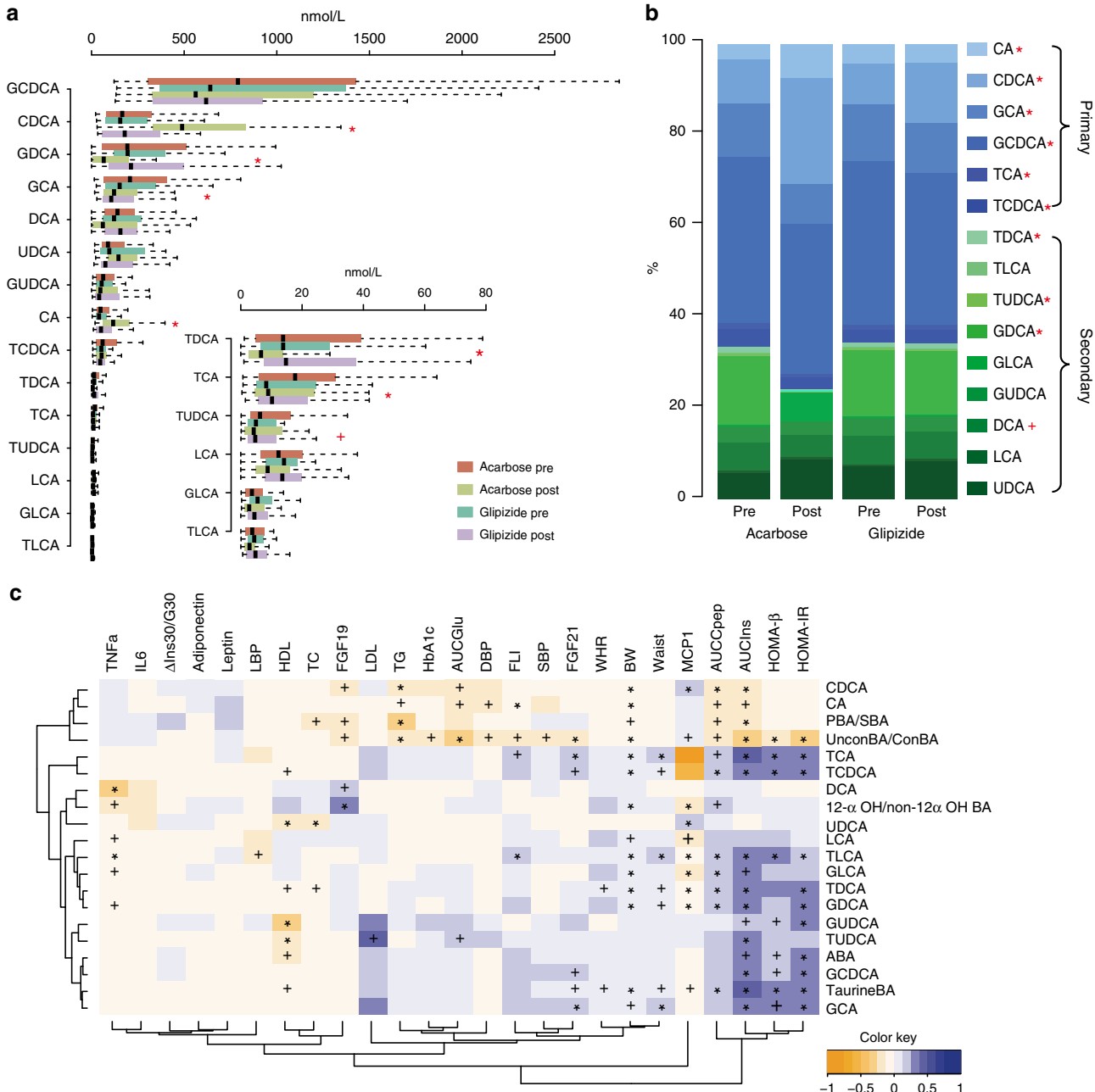

**Fig. 2** Changes in bile acid composition in response to Acarbose treatment correlate with clinical outcomes. **a** Plasma levels of bile acids (BAs) before and after Acarbose and Glipizide treatments. Results are shown as boxes denoting the interquartile range between the first and third quartiles. The line within the boxes denotes the median, paired Wilcoxon rank-sum test, *$P < 0.01$, +$P < 0.05$: Post-treatment vs. Pre-treatment in the Acarbose arm, $n = 49$ in Acarbose pre and Acarbose post-treatment, $n = 43$ in Glipizide pre and Glipizide post-treatment. **b** Changes in the composition of BAs in the two treatment arms; paired Wilcoxon rank-sum test, *$P < 0.01$, +$P < 0.05$: Post vs. Pre-treatment in the Acarbose arm. **c** Multivariate GEE analysis of the contribution of BAs to the main clinical outcomes of Acarbose treatment. *$P < 0.01$, +$P < 0.05$, adjusted for BMI, sex and age. The colour key represents the regression coefficients of the independent variables. AUCIns, Area under curve value of plasma insulin level during a meal test; AUCCpep, Area under curve value of plasma C peptide level during a meal test; AUCGlu, Area under curve value of blood glucose level during a meal test; FLI, fatty liver index; LBP, lipopolysaccharides binding protein; ΔIns30/G30 = (Ins30-Ins0)/(G30-G0) during a meal test. TC total cholesterol, LDL low-density of lipoprotein cholesterol, TG triglycerides, HDL high-density of lipoprotein cholesterol, IL6 Interleukin 6, CDCA chenodeoxycholic acid, CA cholic acid, PBA/SBA the ratio of primary BAs vs. secondary BAs, UnconBA/ConBA the ratio of unconjugated BA species vs. conjugated BA species, TCA taurocholic acid, TCDCA taurochenodeoxycholic acid, DCA deoxycholic acid, 12αOH/non-12αOH the ratio of 12αOH vs. non-12αOH BA species, UDCA ursodeoxycholic acid, LCA lithocholic acid, TLCA taurolithocholic acid, GLCA glycolithocholic acid, TDCA taurodeoxycholic acid, GDCA glycodeoxycholic acid, GUDCA glycoursodeoxycholic acid, TUDCA tauroursodeoxycholic acid, ABA sum of all measured plasma BAs, GCDCA glycochenodeoxycholic acid, TaurineBA taurine-conjugated bile acids, GCA glycocholic acid

rank-sum test, $P > 0.05$ between groups, Fig. 1a, b). However, the reductions in body weight (BW) and BMI were more pronounced in the Acarbose arm than in the Glipizide arm after 3 months of treatment (Fig. 1c). The area under the curve (AUC) value of insulin release was significantly reduced after Acarbose treatment, but increased as expected after Glipizide treatment during a carbohydrate meal test containing 100 g flour as described previously[24], (1029.9 ± 575.2 pmol/Lxmin vs. 637.1 ± 342.6 pmol/Lxmin, Acarbose pre vs. post-treatment, $P = 2.41E-06$; 940.2 ± 431.7 pmol/Lxmin vs.1139.6 ± 608.9 pmol/Lxmin, Glipizide pre vs. post-treatment, $P = 0.019$; paired Wilcoxon rank-sum test; Fig. 1d, e). In addition, patients receiving Acarbose but not

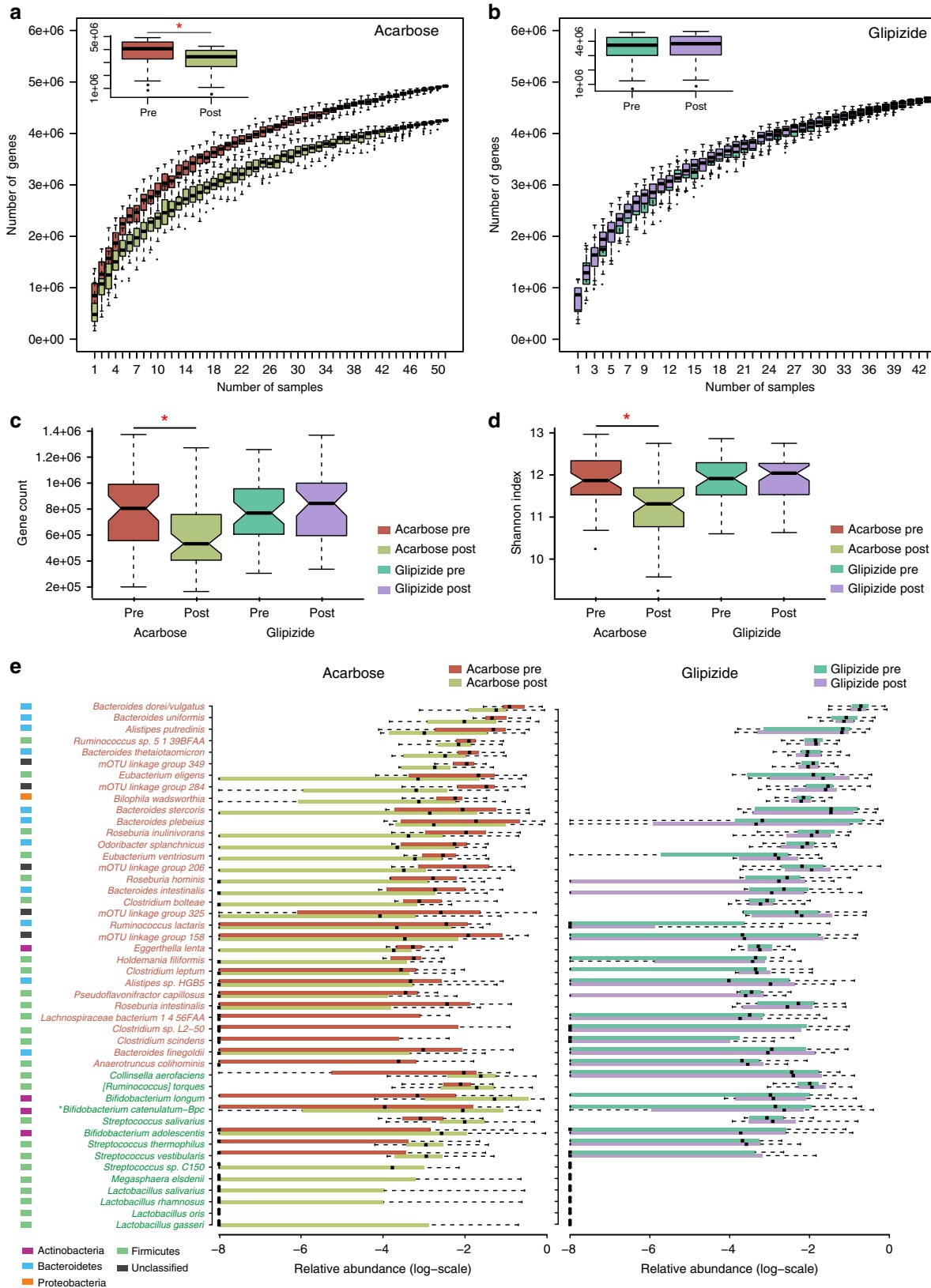

Glipizide showed a significant improvement in clinical parameters that are major risk factors for metabolic comorbidities and cardiovascular complications of T2D, such as the homeostasis model assessment of insulin resistance (HOMA-IR), total cholesterol (TC), triglyceride (TG) and fatty liver index (FLI)[25]. Of note, neither Acarbose nor Glipizide treatment increased, but rather decreased plasma FGF19 levels, a key factor produced in the intestine and shown to be of importance for metabolic health[26], indicating that improvement in HbA1c and FBG and PBG levels was not dependent on FGF19, (Table 1; Fig. 1f, g, all clinical parameters are given in Supplementary Table 2). Considering the known Acarbose-induced reduction of carbohydrate absorption in the small intestine, we next investigated to what extent Acarbose-induced changes in the gut microbiota might contribute to the observed additional metabolic benefits besides the hypoglycaemic effect.

**Changes in plasma BAs correlate with therapeutic benefits.** Alterations in the composition of the BA pool in humans can reflect intraluminal changes in the gut modulating gut-derived signals that regulate host metabolism and alteration in plasma BAs have been linked with metabolic disorders[12]. Thus, the level of taurine-conjugated BAs (TaurineBA, the sum of all taurine-conjugated BAs) is elevated in T2D[15] and the ratio of 12-α OH to non-12α OH BAs (12-α OH/non-12α OH BA ratio) in the total BA pool has been linked to insulin resistance[14]. In our study, we assayed 15 BA species using high-performance LC-MS/MS on fasting plasma samples. At baseline, no differences in plasma BA levels were observed between the treatment arms (Fig. 2a). After treatment, plasma BA composition was substantially altered solely in the Acarbose arm with the level of total BAs remaining unchanged (Fig. 2b, Supplementary Table 3, paired Wilcoxon rank-sum test, P< 0.05). Acarbose treatment decreased the levels of plasma secondary BAs (SBAs), mainly conjugated deoxycholic acids (DCAs). Due to the increased levels of cholic acid (CA) and chenodeoxycholic acid (CDCA), we observed an increased overall level of unconjugated primary BAs (PBAs). Further, the ratios of PBAs to SBAs (PBA/SBA ratio) and unconjugated to conjugated BAs (UnconBA/ConBA ratio) increased, whereas the levels of TaurineBA and the 12-α OH/non-12α OH BA ratio decreased after Acarbose treatment (Supplementary Table 3, paired Wilcoxon rank-sum test, P < 0.05).

A multivariate correlation analysis based on the generalised estimated equation (GEE) revealed that the Acarbose-induced improvements of glycaemic control and other metabolic parameters were correlated with the changes in plasma BA composition (Fig. 2c; Supplementary Table 4). The significant increases in the UnconBA/ConBA ratio and PBA/SBA ratio correlated with improvements in HbA1c, BW, FLI, blood pressure and lipid profile (Fig. 2c, Supplementary Table 4). Via deconjugation and the sequential dehydroxylation of PBAs to produce SBAs, the gut microbiome is crucial for regulating BA diversity[27]. The increased UnconBA/ConBA and PBA/SBA ratios

after Acarbose treatment indicated that Acarbose treatment might alter two key steps of bacterial BA biotransformation.

**Acarbose modulates the composition of the gut microbiota.** We next performed whole-metagenome sequencing on the 188-paired faecal samples from the patients and generated 1.42 Tb of high-quality sequencing data with an average of 7.7 Gb per sample to investigate the differential responses of the gut microbiome to the two types of antidiabetic treatment. Genes were identified by aligning the reads to the 9.9 M integrated reference catalogue of genes in the human gut microbiome[28]. On average, 80.5% of the reads in each sample were mapped (Supplementary Table 5). Metagenomic analyses demonstrated that Acarbose treatment significantly altered the relative abundances of 141,382 microbial genes (paired Wilcoxon rank-sum test, q < 0.01), whereas no significant changes in relative gene abundances were observed following Glipizide treatment (Supplementary Table 5). Rarefaction analyses revealed that the number of genes in the microbiomes of patients in the Acarbose arm surprisingly was significantly decreased, whereas no changes were observed in the Glipizide arm (Fig. 3a, b). In line with this observation, both gene count and Shannon index decreased after Acarbose treatment, but not after Glipizide treatment (Fig. 3c, d). Analysis of species-level molecular operational taxonomic unit (mOTU) profiles demonstrated that Acarbose led to significant changes in the relative abundances of 69 mOTUs (paired Wilcoxon rank-sum test, q < 0.01; Fig. 3e; Supplementary Table 6); Glipizide treatment did not affect the relative abundances of any mOTUs at this significance level. At baseline, there were no significant differences in relation to gene rarefaction, gene richness and gut microbiome alpha diversity or taxonomic richness between the two arms (Supplementary Fig. 2, Fig. 3c, d). Of note, despite of the comparable hypoglycaemic effect of Glipizide and Acarbose treatment, the gut microbiome alteration after Glipizide treatment was small, suggesting that changes in blood glucose levels were not a primary driver of the changes in the distal gut microbiota. Unlike Glipizide that regulates the ATP-sensitive potassium (K_ATP) channel activity of pancreatic β cells, Acarbose impedes the hydrolysis and absorption of carbohydrates in the small intestine, which in turn changes the abundances of substrates for fermentation by the microbiota in the distal intestine, thereby allowing saccharolytic *Lactobacillus* and *Bifidobacterium* species to thrive and depleting the original distal gut-residing, putrefactive species of *Bacteroides*, *Alistipes* and *Clostridium* (Fig. 3e, paired Wilcoxon rank-sum test, q < 0.01).

**Plasma BA changes, metabolic benefits and gut microbiota.** A mutual interaction between host BAs and the gut microbiome modulates the microbial biotransformation of BAs[29]. In the untreated T2D patients at baseline, permutation multivariate analysis of variance (PERMANOVA) analysis revealed that plasma levels of the SBAs, ursodeoxycholic acid (UDCA), glycoursodeoxycholic acid (GUDCA), lithocholic acid (LCA) and

**Fig. 3** Acarbose elicits stronger impact on the gut microbiota than Glipizide. Effects of Acarbose **a** and Glipizide **b** treatments on gene rarefaction, n = 51 in Acarbose pre and Acarbose post-treatment, n = 43 in Glipizide pre-treatment and Glipizide post-treatment. All bar plots are shown as mean ± S.D. The number of genes in each group was calculated after 100 random samplings with replacement. The effects of treatment in both arms on gene richness **c** and Shannon index **d** of the gut microbiome, n = 51 in Acarbose pre and Acarbose post-treatment, n = 43 in Glipizide pre and Glipizide post-treatment. **e** mOTUs significantly changed in abundance by Acarbose treatment (left panel), Benjamini–Hochberg q-value < 0.01, presented in box-plots to illustrate the relative abundances of the different groups. The mOTUs are listed in the order of their relative abundances pre-treatment. mOTUs in red represent mOTUs that decreased in abundance are after Acarbose treatment, whereas mOTUs that increased in abundance are given in green. The phylum colour code is indicated to the left of the mOTU annotation. The same mOTUs did not change after Glipizide treatment (right panel of **e**). All plotted boxes are interquartile ranges. Dark lines in the boxes indicate medians, the lowest and highest values within 1.5 times IQR from the first and third quartiles. Outliers are shown as circles beyond the whiskers. n = 51 in Acarbose, n = 43 in Glipizide, *Bifidobacterium catenulatum—Bpc: Bifidobacterium catenulatum-Bifidobacterium pseudocatenulatum complex

DCA correlated with changes in the composition of the gut microbiota (Bray–Curtis distance, $q < 0.01$; Supplementary Table 7). Interestingly, 58 out of the 69 mOTUs that exhibited significant changes in relative abundance in response to Acarbose treatment also correlated with Acarbose-induced alterations in plasma BA composition (GEE adjusted for age, sex and BMI, $q < 0.05$; Supplementary Data 1), pointing to a tight link between Acarbose-dependent alterations of the gut microbiota and plasma BA composition.

Many of the Acarbose-induced species have been classified and used as probiotics, and the increase in the relative abundance correlated with improvements in metabolic parameters (GEE adjusted for age, sex and BMI, $q < 0.05$,; Supplementary Fig. 3). Thus, Acarbose-induced increases in *Lactobacillus gasseri* and *Bifidobacterium longum* correlated inversely with changes in BW and HbA1c. These correlations are consistent with those observed in mice treated with VSL3# (a probiotic supplement containing strains of *Lactobacillus*, *Bifidobacterium* and *Streptococcus*)[30] and

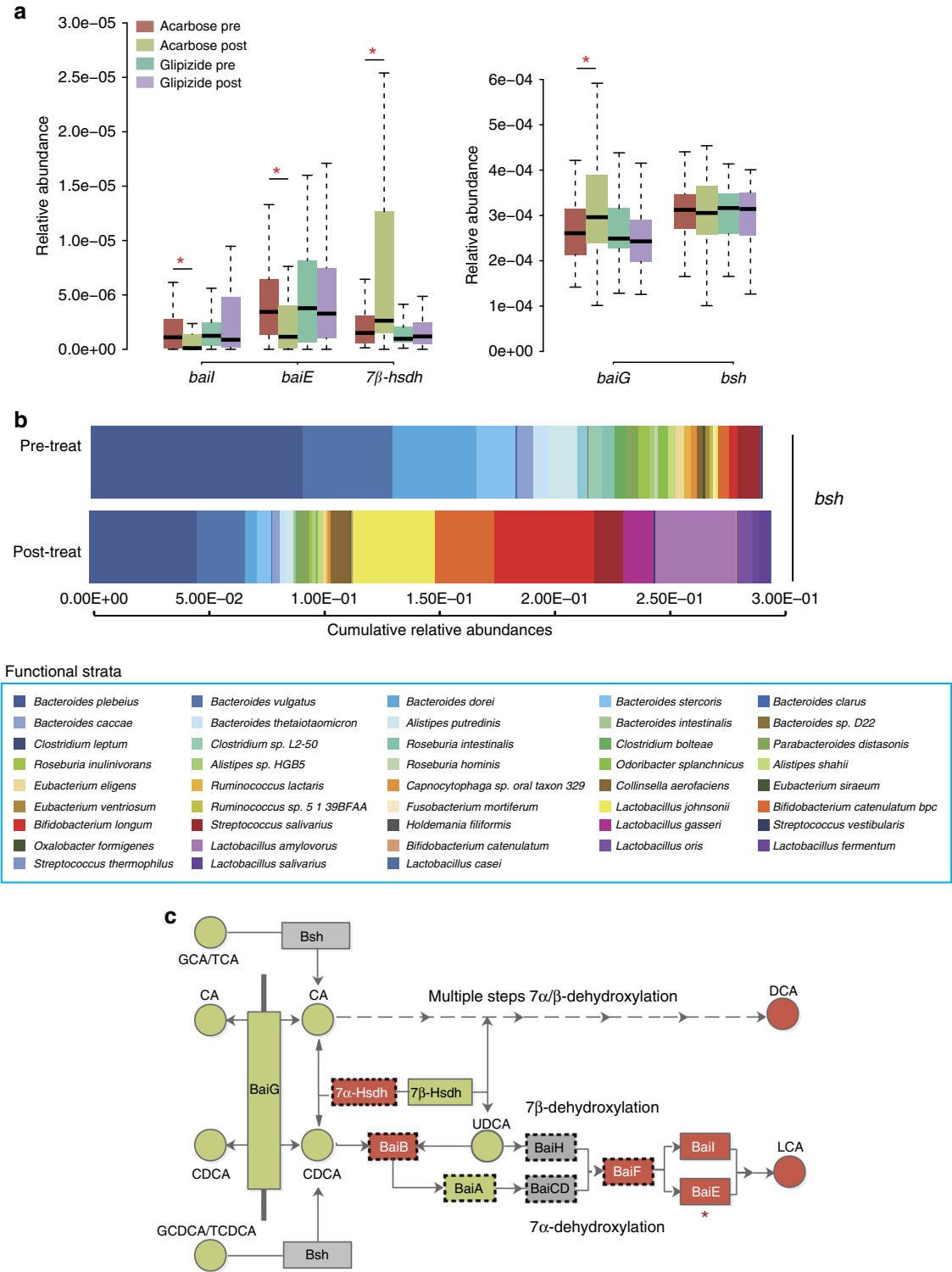

with the effect of microbial overexpression of the gene encoding bile salt hydrolase (Bsh) which also increased the UnconBA/ConBA ratio in mice[31]. The reduced plasma DCA levels and reduced BW gain after Acarbose treatment, correlated with the reductions of Acarbose-depleted species, such as *Bacteroides plebeius*, *Bacteroides dorei/vulgatus* and *Clostridium bolteae*. These species harbour a multi-gene BA-inducible (*bai*) operon encoding genes involved in 7-α/β dehydroxylation of BAs, reported not to be present in probiotic bacteria[27, 32].

**Acarbose treatment alters gut bile acid composition.** To explore the possible impact of Acarbose on gut bacterial BA metabolism, we performed functional annotations of genes using a manually built reference library which includes 1011 annotated amino acid sequences mapped to 16,102 genes implicated in bacterial BA metabolism (Supplementary Table 8). The relative abundances of genes encoding enzymes involved in bacterial BA modification, including BA deconjugation, BA transport, hydroxysteroid dehydrogenation and 7α/β dehydroxylation were thus calculated and compared between pre-treatment and post-treatment samples (Supplementary Table 8). The relative abundances of *baiE* (encoding BA 7α-dehydratase, the rate-limiting enzyme for 7α-dehydroxylation) and *baiI* (encoding BA 7β-dehydratase) were significantly decreased after Acarbose treatment, while the *baiG* (encoding BA transporter) and genes encoding 7β-Hsdh (*7β-hsdh*) showed significant enrichment after Acarbose treatment (paired Wilcoxon rank-sum test, $P < 0.05$; Fig. 4a; Supplementary Table 9). None of these genes changed significantly in relative abundance after Glipizide treatment (paired Wilcoxon rank-sum test, $P > 0.05$; Fig. 4a; Supplementary Table 9).

The taxa distributions of genes involved in BA metabolism (Supplementary Data 2) in baseline samples revealed that *B. plebeius*, *B. vulgatus/dorei* and *B. dorei* largely accounted for the abundances of *bsh*, *baiG* and *baiE*. This suggested that these *Bacteroides* species were mainly responsible for both BA deconjugation and subsequent dehydroxylation in the distal gut microbiota of treatment-naive T2D patients. After treatment, the relative abundances of these genes were reduced due to the loss of their dominant contributor species, or the dominant contributors of some genes were replaced by other species. For example, the dominant contributors of the genes encoding Bsh changed from *B. plebeius* and *B. vulgatus/dorei* to *B. longum*, and *Lactobacillus amylovorus* (Fig. 4b). The *Bifidobacterium* and *Lactobacillus* genera have been reported to possess high Bsh activity, but are lacking the *bai* genes[32–34].

We investigated if the changes in the bacterial potential for BA metabolism induced by Acarbose treatment were reflected in the BA composition in the faecal samples. The total level of faecal BAs was significantly decreased after Acarbose treatment (Supplementary Fig. 4a, b, Supplementary Table 10). Actually, the faecal levels of the primary BAs, CA and CDCA, increased following Acarbose treatment, but this was more than counteracted by the large decrease in the levels of the secondary BAs, DCA and LCA (Supplementary Fig. 4a). Overall, these changes resulted in an increased faecal PBA/SBA ratio in response to Acarbose treatment (Supplementary Fig. 4d). The faecal UnconBA/conBA ratio was unchanged (Supplementary Fig. 4c). These changes mirrored the changed relative abundance of *bai* genes with an unchanged relative abundance of *bsh* genes following Acarbose treatment (illustrated in Fig. 4c), but do not explain the significant increase in the plasma UnconBA/ConBA ratio after Acarbose treatment.

In addition to altering BA metabolism, bacterial functional pathway analysis showed that Acarbose treatment also increased the potential for glutamine synthesis, decreased the potential for glycosaminoglycan degradation, and decreased the potential for $H_2S$ and methane production (Supplementary Data 3), all associated with host-gut barrier integrity and T2D risks[3, 35–38]. Finally, both antidiabetic treatments reduced the abundances of genes involved in type III secretion and LPS biosynthesis that are linked to insulin resistance[39, 40], suggesting the existence of certain common routes in the response of the gut microbiome to antidiabetic treatments despite the use of different medical agents.

**Microbiota composition predicts the response to Acarbose.** Studies have suggested a stratification of individuals by clustering algorithms based on the genus composition of the intestinal microbiota[41, 42], but the concept of the existence of discrete enterotypes has also been questioned, suggesting a continuous distribution of enterotypes in an individual[43]. In our study, the baseline microbiomes of patients could be clearly separated into two enterotype-like clusters, one driven by *Bacteroides* (Cluster B) and the other driven by *Prevotella* (Cluster P) shown by PCA analysis (Supplementary Fig. 5a), network analysis (Supplementary Fig. 5b) and abundance box plot (Supplementary Fig. 5c). In agreement with previous reports, anthropomorphic, metabolic and biochemical parameters did not differ between patients from the two clusters[41] (Wilcoxon rank-sum test, $P > 0.05$, Supplementary Table 11). However, already at baseline patients in Cluster B had lower levels of LCA and DCA, and higher levels of UDCA in the BA pool both in plasma and faeces. (Wilcoxon rank-sum test, $P < 0.05$, Fig. 5a; Supplementary Tables 12, 13). Patients in Cluster B also exhibited a higher plasma PBA/SBA ratio and lower 12-α OH/non-12α OH BA ratio compared to

**Fig. 4** Acarbose treatment affects the potential for secondary bile acid metabolism. **a** Comparison of the relative abundances of genes encoding enzymes involved in SBA metabolism exhibiting significant alterations in abundance after Acarbose treatment, but not after Glipizide treatment; Plotted boxes are interquartile ranges. Dark lines in the boxes indicate medians, the lowest and highest values within 1.5 times IQR from the first and third quartiles, paired Wilcoxon rank-sum test, *$p < 0.01$, +$p < 0.05$. $n = 51$ in Acarbose pre and Acarbose post-treatment, $n = 43$ in Glipizide pre and Glipizide post-treatment. *baiG*, *baiI* and *baiE*, bile acid-inducible (*bai*) gene G, I and E; *hsdh*, gene encoding hydroxysteroid dehydrogenase, *bsh*, gene encoding bile salt hydrolase. **b** Cumulative relative abundances of genes encoding Bsh (EC.3.5.1.24) listed according to the contribution by annotated bacterial species pre-Acarbose and post-Acarbose treatment. The *x*-axis represents the cumulative relative abundances of genes encoding Bsh. Only species where the genes encoding Bsh constituted more than 0.5% of the total abundance of these genes, and where we observed a significant difference in relative abundances of these genes in response to Acarbose treatment, are included, $q < 0.05$, paired Wilcoxon rank-sum test. **c** Schematics of the bacterial BA biotransformation pathways, including deconjugation and multiple steps of 7α/β-dehydroxylation. Faecal BAs and BA-metabolising enzymes encoded by genes enriched after Acarbose treatment are marked in green, those depleted after Acarbose treatment are marked in red; those that did not change in abundance by treatment are marked in grey. *indicates the rate-limiting enzyme of 7α-dehydroxylation. Bsh, bile acid hydrolase; Bai G/B/A/I/H/CD/E/F, enzymes encoded by the *bai* operon. GCA glycocholic acid, TCA taurocholic acid, GCDCA glycochenodeoxycholic acid, TCDCA taurochenodeoxycholic acid, CA cholic acid, CDCA chenodeoxycholic acid, UDCA ursodeoxycholic acid, DCA deoxycholic acid, LCA lithocholic acid. Boxes framed by dashed lines represent enzymes where the annotation is ambiguous because of inconsistent functional annotation using BlastP based on the Uniprot database vs. BlastKOALA based on the KEGG database. (See Supplementary Methods, Functional annotation of genes involved in BA synthesis based on BLASTP and BlastKOALA for further details)

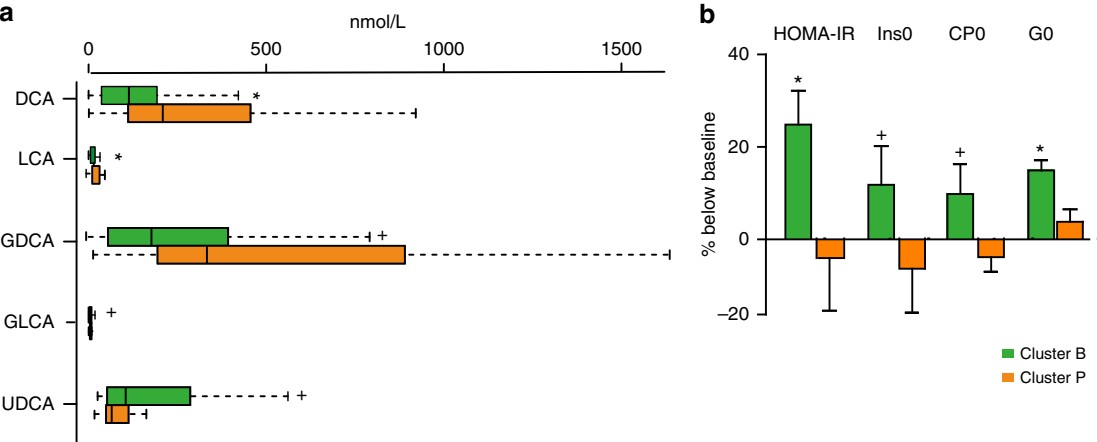

**Fig. 5** Plasma bile acid composition and therapeutic responses to Acarbose treatment differ in patients with a different baseline microbiota composition. **a** Plasma composition of bile acids in patients belonging to baseline enterotype-like clusters driven by *Bacteroides* (Cluster B) or *Prevotella* (Cluster P). Plasma levels of DCA, LCA, GDCA, GLCA and UDCA differed significantly between Cluster B and Cluster P at the baseline; plotted boxes are interquartile ranges. Dark lines in the boxes indicate medians, the lowest and highest values within 1.5 times IQR from the first and third quartiles, Wilcoxon rank-sum test, *$P <$ 0.01, +$P <$ 0.05. DCA deoxycholic acid, LCA lithocholic acid, GDCA glycodeoxycholic acid, GLCA glycolithocholic acid, UDCA ursodeoxycholic acid. **b** Percentage changes over baseline levels of fasting blood glucose (G0), insulin (Ins0), C peptide (CP0) and HOMA-IR differed significantly between Cluster B and Cluster P after Acarbose treatment; paired Wilcoxon rank-sum test, *$P <$ 0.01, +$P <$ 0.05. Cluster B, $n = 36$, Cluster P, $n = 15$ in Acarbose arm, bar plot is shown as mean ± S.D.

patients in Cluster P (Wilcoxon rank-sum test, $P < 0.05$, Fig. 5a; Supplementary Table 12). Further, patients in Cluster B exhibited significantly decreased relative abundance of *baiG*, but increased relative abundances of *bsh*, and *7β-hsdh*, whereas *baiE* and *baiI* exhibited similar relative abundances in both clusters (Wilcoxon rank-sum test, $P < 0.05$, Supplementary Table 14). Interestingly, the clinical effects of Acarbose were more pronounced in patients belonging to Cluster B than in patients belonging to Cluster P. Despite a similar improvement in HbA1c levels by Acarbose treatment, patients in Cluster B exhibited significantly greater improvements in G0 (FBG), insulin, C peptide levels and, therefore, HOMA-IR over baseline levels than patients in Cluster P (Fig. 5b; Supplementary Fig. 6a; Wilcoxon rank-sum test, $P < 0.05$, see also Supplementary Table 15 for all results). We observed the same tendency in the Glipizide arm. The compositions of the gut microbiome of patients in Cluster B changed significantly with depletion of the dominant *Bacteroides* genus and the increase of the *Bifidobacterium* genus after treatment, whereas fewer changes were observed in the gut microbiomes of patients belonging to Cluster P (Supplementary Fig. 6b; Wilcoxon rank-sum test, $P < 0.01$). Further, Acarbose treatment led to a decreased abundance of *baiE* and an increased abundance of *7β-hsdh* and *baiG*, exclusively in patients in Cluster B (Supplementary Fig. 6c). Together, these results suggest that some of the therapeutic effects of Acarbose-induced modulation of the gut microbiota are related to bacterial BA metabolism and, therefore, BA signalling.

## Discussion
This study compared the effects of two antidiabetic drugs on the gut microbiota, plasma BAs and their relations with clinical outcomes in a clinical trial-based cohort. A key finding of this study is that the baseline composition of the gut microbiota may be used for stratification of patients prior to initiation of treatment.

The significant reductions in BW with targeted glycaemic control after Acarbose treatment is in line with earlier studies[44, 45]. Further, we demonstrate that patients receiving Acarbose, but not Glipizide showed a significant improvement in the HOMA-IR, lipid profile and FLI. The results of plasma BA profiling and the multiple correlations between changes of plasma BAs and clinical parameters, including BW, HOMA-IR, lipid profile and FLI, indicate that the modulation of BA signalling may contribute to the unique metabolic effects of Acarbose. The increased levels of plasma CDCA and an increased UnconBA/ConBA ratio induced by Acarbose may enhance in vivo FXR activity[46] and change the faecal BA composition, resulting in decreased intestinal lipid absorption and reduced hyperglycaemia[47, 48]. Interestingly, plasma UnconBA/ConBA and PBA/SBA ratios were also shown to increase after biliopancreatic diversion surgery[13], which in a 2-year follow-up study was reported to result in superior improvements in hyperglycaemia and dyslipidaemia compared to the Roux-en-Y bypass and conventional medical therapy[49]. The notion that similar changes in BA composition and correlations between metabolic traits and plasma BAs after both Acarbose treatment and gut reconstruction surgeries, despite differences in populations, sample size and follow-up time, points to the importance of BA signalling in metabolic control in T2D treatments.

Acarbose treatment led to increased abundances of species possessing high Bsh activity, mainly *B. longum*, *L. gasseri* and reduced abundance of the SBA producers, *B. plebeius*, *B. vulgatus/dorei* and *C. bolteae*. Further functional analysis showed decreased abundances of genes encoding the rate limiting enzyme for 7α/β dehydroxylation and a switch in the main contributors of BA deconjugation enzymes from *Bacteroides* spp. to *Lactobacillus* and *Bifidobacterium* spp., which barely contain *bai* genes. Thus, the metagenomics analyses indicated lower capacity for BA 7α/β dehydroxylation after Acarbose treatment, which is in keeping with the observed increased plasma and faecal PBA/SBA ratios.

Interestingly, the increased ratio of UnconBA/ConBA was observed in plasma, but not in faecal samples, and the abundances of bacterial *bsh* genes did not change either. This would argue against an increased potential of bacterial deconjugation in response to Acarbose treatment, even in patients harbouring higher abundancies of species such as *B. longum* and *L. gasseri* which have been reported to possess high Bsh activity. However, deconjugation occurs in the upper part of intestine and most of unconjugated PBAs are reabsorbed in the ileum, and thus, to

what extent the UnconBA/ConBA ratio in the faecal sample represents a valid proxy for in vivo bacterial deconjugation capacity needs to be further investigated. It is possible that the significantly reduced level of FGF19, which conveys the main inhibitory intestinal BA signal for hepatic BA neo-synthesis[30], [50], [51] or an altered intestinal BA reabsorption after Acarbose treatment may contribute to the discrepancy between the UnconBA/ConBA ratios in plasma and faecal samples. However, taken together our results clearly point to the importance of Acarbose-induced changes in the composition of the gut microbiome and the observed alterations in BA profiles.

With the increased attention on the metabolic significance of BA signalling, the discovery of an Acarbose-gut microbiota-BA axis may aid the identification of novel modalities for therapeutics intervention targeting the gut microbiome or plasma BAs. The implication of gut microbiota-plasma BA axis underlying therapeutic effects of Acarbose is further supported by the finding of more pronounced changes in plasma BA composition and clinical benefits of Acarbose treatment in patients where the baseline microbiomes belonged to an enterotype-like cluster driven by *Bacteroides* (Cluster B) vs. those belonging to an enterotype-like cluster driven by *Prevotella* (Cluster P).

In conclusion, our results suggest that the differential therapeutic responses are related to distinct abilities of the microbial communities in the two microbiome clusters to metabolise BAs, and, more importantly, our findings demonstrate that stratification of patients according to baseline gut microbiota composition may provide a tool for selecting medication strategy and predicting antidiabetic metabolic benefits.

## Methods

**Trial design**. This study was a randomised, open-label, two-arm, multicentre clinical trial. Five centres participated in this study: Ruijin Hospital, Renji Hospital, Xinhua Hospital, all affiliated to Shanghai Jiaotong University Medical School, Shanghai General Hospital affiliated to Shanghai Jiaotong University and Central Hospital of Minhang District affiliated to Fudan University Medical School. The study was approved by the Institutional Review Board at each hospital and is registered at ClinicalTrials.gov (NCT01758471). Written informed consent was obtained from each patient. The study was conducted in accordance with the principles of the Declaration of Helsinki.

**Randomisation and treatment**. After administration of a glucose tolerance test and evaluation of other baseline characteristics, eligible patients were diagnosed according to the 1999 World Health Organization criteria with fasting plasma glucose ≥7 mmol/L and/or an oral glucose tolerance test (OGTT) 2 h≥11.1 mmol/L[52] and randomly assigned to receive either Acarbose or Glipizide treatment, (detailed screening and recruitment procedures are provided in the Supplementary Methods, Screening, recruitment, dietary information and follow-up). The randomisation codes were generated with SAS (version 9.2; SAS Institute, Cary, NC) by a biostatistician who did not participate in the enrolment of patients. A formal power analysis was not performed to calculate sample size. The sample size for this interventional study was selected based on previous studies of changes in the composition of the gut microbiota in T2D[3] and other diseases that compared the effects of different therapeutic strategies on intestinal microbiota[53], [54], considering potential withdraw and loss to follow-up.

Patients were instructed to visit every month for the measurement of fasting and postprandial glucose levels. The drug dosage was modified according to the glycaemic control at each visit (targeted FBG < 7.0 mmol/L and PBG < 10.0 mmol/L). After 3 months of treatment, all patients were again confined in the hospital as for the run-in period, served the same meals for lunch and dinner as during the first visit, fasted overnight and given an oral glucose load the next morning. Blood samples, faeces and urine were collected as described above (more details are provided in the Supplementary Methods, Screening, recruitment, dietary information and follow-up).

**Sample preparation**. One day before the first visit, the enrolled patients were confined in the hospital and served the same meals for lunch and dinner. Faeces and urine were collected in dispensable sterile containers. Samples were kept on dry ice before being stored at −80°C. Patients were fasted overnight. At 0800 hours, the patients ingested a carbohydrate meal containing 100 g of flour as an oral glucose load[24]. Blood samples were taken before and 30 min, 60 min, 120 min and 180 min after the meal test. The blood samples were immediately supplemented with a dipeptidyl peptidase-4 inhibitor (for GLP-1 measurement; Millipore, Billerica, MA,

USA), the serine protease inhibitor Pefabloc SC (AEBSF for active ghrelin measurement; Roche, Basel, Switzerland) and a protease inhibitor cocktail (Millipore, Billerica, MA, USA). Serum and plasma samples were prepared and kept on dry ice before storage at −80°C.

**Laboratory assays**. Plasma levels of cytokines and hormones were measured using standard kits (Millipore, Billerica, MA, USA) on a Luminex FlexM3D™ system with xMAP Luminex technology. HbA1c was measured by high-performance liquid chromatography using the VARIANT II HbA1C Testing System (Bio-Rad Laboratories, Hercules, CA, USA). Plasma BAs were assayed by LC-MS/MS with multiple reaction monitoring, using a 1290 Infinity LC system (Agilent, Santa Clara, CA, USA) coupled with 6460 A Triple Quadrupole mass spectrometry (Agilent, Santa Clara, CA, USA), and the Waters Acquity UPLC system (Waters, Milford, MA) coupled to a Triple Quad™ 5500 tandem mass spectrometer (AB Sciex, Framingham, MA) was used in analysis of BAs in faecal samples. Chromatographic separation was performed on a 100 mm × 2.1 mm ACQUITY UPLC C8 column for plasma samples and UPLC BEH C18 column for faecal samples with 1.7 μm particle size (Waters, Milford, MA, USA). Sample preparations for BA assay were performed as described previouly[55].

Glucose and biochemical parameters for assaying hepatic function, renal function, blood electrolytes and lipid profiles were determined using the AU5800 Clinical Chemistry System (Beckman Coulter, Brea, CA, USA).

**Metagenomic sequencing and annotation**. DNA was extracted from stool samples as described previously[3]. We constructed one DNA paired-end (PE) library with an insert size of 350 base pairs (bp) for each sample following the manufacturer's instructions (Illumina, San Diego, California, USA). All samples were sequenced using the 100-bp PE strategy on Illumina HiSeq 2500. The sequencing data are available at the European Nucleotide Archive with code PRJEB12124. Adaptor contamination and low-quality reads were discarded from the raw reads, and the remaining reads were filtered to eliminate human host DNA based on the human genome reference (hg18). The gene profiles were generated by aligning high-quality sequencing reads to the reference gene catalogue as previously described[28]. Similarly, the relative abundance profiles of genera, species and Kyoto Encyclopaedia of Genes and Genomes (KEGG) orthologous groups (KOs) were determined from the gene abundances of mOTUs were obtained using mOTU profiling software[56].

**Analysis of richness and biodiversity**. Rarefaction and α-diversity analysis (within-sample diversity) were performed as described by Li et al[27]. The gene counts in each faecal sample were determined as previously reported[57]. Rarefaction analysis was performed to assess gene richness in the T2D patients before and after the patients had been treated with drugs for 3 months. For the rarefaction analysis, we performed random sampling 100 times in the cohort with replacement and estimated the total number of genes that could be identified from these samples using the Chao2 richness estimator. To minimise erroneous identification, only the genes with ≥1 pair of mapped reads were considered to be present in a sample.

**Correlations between microbial genes and plasma BA species**. The interactions between gene abundance profiles and each of the 15 plasma BA species were assessed by PERMANOVA with the Bray–Curtis distance. The number of permutations was 9999 (performed using the vegan package in R 3.3.1).

**Multivariate regression analysis**. A GEE analysis (corrected for age, sex and BMI) was performed to explore Acarbose-induced longitudinal associations between changes in (1) BA species and clinical parameters, (2) BA species and the mOTUs significantly altered in relative abundance by Acarbose and (3) the mOTUs significantly altered in relative abundance by Acarbose and clinical parameters. The P value of each regression coefficient was calculated, and an adjusted P value of <0.05 was considered significant (performed using the gee package in R 3.3.1).

All regression analyses between BW and other traits were conducted after adjustment for age and sex, but not BMI.

**KEGG enrichment analysis**. Differentially enriched KO modules were identified according to their reporter scores from the Z-scores of individual KOs as described previously[58]. An absolute value of reporter score of 1.95 or higher was used as the detection threshold for significance.

**Clustering of microbial community types**. Samples were clustered based on the genus profiles. Two enterotype-like clusters were identified in our samples using the method described previously[36]. In this study, samples were clustered using Jensen–Shannon distance and presented by a PCA (Principal Component Analysis) graph implemented in the 'ade4' package in R.

**Identification of microbial genes involved in BA biotransformation**. We retrieved 11 enzymes involved in microbial BA metabolism from UniProt (Universal Protein Resource) protein database. After removing redundant sequences with greater than 90% identity, a total of 1011 amino acid sequences were obtained

and used as the reference database for the blast search. Detailed information, including the Enzyme Commission number (EC number), KO number, protein name and search keywords are provided in Supplementary Table 8.

The 1011 amino acid sequences were then aligned with the predicted amino acid sequences translated from the 9.9 M gene catalogue using NCBI-BLASTP (E-value < 1E−5 and bit score > 60). In total, 16,102 genes potentially encoding enzymes involved in BA metabolism were identified. Additionally, we assessed the 16,102 potential-hits against KEGG using BlastKOALA (KEGG Orthology And Links Annotation) algorithm[59] and compared the best hits with UniProt based BLASTP results. Accordingly, only genes exhibiting highly consistent annotation (bsh, baiE, baiG, baiI and 7β-hsdh) have been included in the analyses (for more detailed description and discussion, see Supplementary Methods, Functional annotation of genes involved in BA synthesis based on BLASTP and BlastKOALA, and Supplementary Data 4–7).

To identify dominant species harbouring BA-metabolising enzymes, we summed the total relative abundances of genes with a potential to encode enzymes involved in BA metabolism and calculated the cumulative relative abundances of these genes according to the contribution of each species harbouring the genes: (relative gene X abundance in one species/total relative abundances of gene X) × 100%.

If the species contributed more than 0.5% of the total pool, this species was considered as a dominant contributor.

**Statistical analyses**. Paired Wilcoxon rank-sum tests were conducted to detect differences in clinical parameters, BAs, genes and mOTUs pre-treatment and post-treatment. P values were adjusted by the Benjamini–Hochberg correction. A P value or an adjusted P value of less than 0.05 was considered significant.

**Data availability**. Metagenomic sequencing data for all samples have been deposited in the European Bioinformatics Institute (EBI) database under accession code PRJEB12124. All other data are available upon request.

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

## Acknowledgements

We thank Mr. Yixin Wu, Mr. Wenzhong Zhou and Ms. Qun Zhang of the Central Laboratory, Shanghai Institute of Endocrine and Metabolic Diseases, for help with analysing HbA1c levels. We also thank Dr. H. Bjørn Nielsen and Dr. Susanne Brix of the Technical University of Denmark for their valuable inputs and discussions. We also thank Shanghai ProfLeader Biotech Co., Ltd. for the assistance with the analysis of faecal BAs. This work was supported by grants from the National Basic Research Program of China (2015CB5536003), Bayer HealthCare, the National Nature Science Foundation of China (81621061, 81670761, 81370963), National International Science Cooperation Foundation (no. 2015DFA30560), Shanghai Rising-Star Program (17QA1403300), Shanghai Municipal Education Commission-Gaofeng Clinical Medicine Grant (No. 20161411), Shanghai Municipal Education Commission and Shanghai Education Development Foundation (No. 14SG17), Three year Action Plan and Public Health, Phase IV, Shanghai (No.15GWZK0802), National Key R&D program of China (2016YFC1305601) and Shenzhen Municipal Government of China (JSGG20160229172752028 and CXB201108250098A).

## Author contributions

Y.G., G.N., W.W., Q.F., Y.Z. and R.L. designed the study. W.W., Y.G., Y.Z., J.H., W.G., Y.B., X.Xie, W.L., J.M., Q.S., H.Zha., Y.P. and J.Y. enrolled and followed-up patients. Y.G., Y.Z., X.Xie, J.H., W.G. Y.B. and W.W. performed the clinical part of the study. Y.G., Y.Z., X.Xie, H.Zho., J.L., R.L., J.H., H.R., W.G., Y.B., W.W. and G.N., analysed the clinical data. Xiaokai.W., J.L., H.Zho., D.Z., Q.F., H.R., H.X., Xu.X. and Y.H. performed the metagenomic sequencing part of the study. Xiaokai.W., H.Zho., J.L., D.Z., Q.F., H.R., F.L., Xi.X., H.X., Y.G., R.L., Xu.X., Y.H., L.M. and K.K. analysed metagenomic sequencing data. Xiaolin.W., X.Z. and G.X. performed the LC-MS/MS analyses of plasma BA profile. Y.G. Xiaokai.W., J.L., H.Zho., H.R., D.Z., Q.F., correlate different omics data. Y.G. Xiaokai.W., Y.Z., J.L., H.Zho., G.N., W.W., D.Z., Q.F., Xu.X., L.M. and K.K. wrote the manuscript. Y.G., Xiaokai.W., J.L., Y.Z., R.L., H.Zho., D.Z., Q.F., Xu.X., Y.H., J.H., Y.B., W.G., K.K., L.M., W.W. and G.N. participated in discussions.

## Additional information

**Competing interests:** The authors declare no competing financial interests.

