## [Peer Review File · Nature Communications]

Reviewers' comments:

Reviewer #1 (Remarks to the Author):

This paper by Gu et al describes the effect of acarbose vs glipizide on bile acid metabolism (fasting plasma samples) between baseline and 3 months after start treatment in 106 naive chinese DM2 patients (1:1 randomized, 94 of the enrolled patients completed the study 51 to acarbose and 43 to glipizide).. Acarbose decreased plasma secondary bile acid and increased unconjugated primary bilacids. Moreover, several correlations between bilacids and elated with improvements in HbA1c, BW, FLI, blood pressure and lipid profiles were observed. Only acarbose treatment significantly affected fecal microbiota composition (predominantly saccharolytic Lactobacillus and Bifidobacterium species) that were also related to bilacid levels in plasma. Subsequent functional analyses showed that 7-alfa-hydroxysteroid dehydrogenase were significantly decreased in microbial genes after Acarbose, while the baiG (encoding bile acid transporter) and genes encoding 7-beta-HSDH were enriched upon acarbose; these were all related to B. plebeius, B. vulgatus/dorei and B. dorei. These findings were aligned by clustering of enterotypes with success of acarbose treatment (bacteroides cluster > prevotella cluster).

Overall this is an interesting study, however several major comments preclude publication of this paper and need to be addressed since they can induce major bias on the microbiota analyses and outcome.

1. can the authors provide Dietary data during the study? This as diet is known to affect both microbiota composition (David, Nature 2014) and bileacids excretion (O'Keefe SJ, Nat Comm 2015 :6342)
2. Did the authors also determine changes in Fecal bilacids, if not can they provide these data. This as this would largely add to the understanding of which bacterial strains are involved in altered bileacids metabolism upon acarbose treatment.
3. since predominantly B. plebeius, B. vulgatus/dorei and B. dorei, this suggests that SCFA like butyrate are also involved Can the authors show SCFA levels in feces before and after treatment? can the authors provide these data to give more insight into mode of mechanism?
4. Enterotypes quite controversial and debated in the field (Knights, Cell Host Microbe. 2014) . It would be advisable to add such a reference this to the limitation section of this paper.

Reviewer #2 (Remarks to the Author):

Gu et al. examined the response of gut microbiota and plasma BAs to the anti-diabetic drugs Acarbose and Glipizide in naïve type 2 diabetic (T2D) patients. The authors report that Acarbose specifically, and not Glipizide, increased unconjugated BAs and the primary/secondary BA ratio, increased Lactobacillus and Bifidobacterium abundance, whilst decreasing Bacteroides abundance. The authors observed that these changes were dependent on the host microbiome at baseline prior to treatment. Patients with baseline Bacteroides enterotype exhibited an improved response to Acarbose than those with Prevotella. These findings suggest that baseline microbiota enterotype may be used as a stratification for anti-diabetic treatments.

Whilst the suggestion of stratification of patients based on enterotype to predict response to anti-diabetic treatment is an intriguing possibility, there are some concerns with the statistical data analysis, cluster analysis and the presentation of the data.

Comments:

1. Patients recruited from 5 centers. Was this taken into account when measuring the baseline

enterotypes. PCA analysis of enterotype of all patients to ensure no segregation based on center recruitment.

2. Standard units for lipid measurements are mg/dL. Graphs in Figure 1 should be changed to reflect these units. The normal range for healthy and naïve T2D should also be indicated.

3. The presentation of the data in Figure 1d is extremely confusing. It would be more appropriate to have separate graphs for Acarbose and Glipizide.

4. There is no label on the y-axis of the HOMA-IR graph in Figure 1f.

5. The data of Table 1 show that the effect of Acarbose are greater than the effects of Glipizide. However, the magnitude of this effect is the same in baseline measurements, which are stated as being not significant. Additionally, a difference of 2kg with an SEM of 10 seems unlikely to be significant. Are the data really non-parametric? More information about the statistical analysis is needed, and preferably plots of the individual data points should be included.

6. The BA data in Table 2 is extremely confusing. Considering the total BA concentrations do not change, but the BA pool composition is significantly altered, a more appropriate way to display this data is either in box plot or pie chart.

7. There is no label on the y-axis of Figure 3a. Is this gene number? Or abundance? The reference in the text would indicate gene number.

8. What are the different colored bars in Figure 3e? There is no reference in the text, and no legend on the graph.

9. What do the two colors of mOTU indicate? Are they ranked in some way? The boxes to the left indicate the bacterial phyla, but it is not clear what this other separate segregation is.

10. In the cluster analysis, was the distribution of cluster B and cluster P equally distributed throughout the 5 centers? Was this taken into account during the cluster analysis?

11. Some references are not correct. For example, references 4, 5 and 8 are used as citations for the effect of the microbiome post gastric bypass surgery. However, the data in these citations states that reductions in diabetes post gastric bypass could not always be explained by weight loss but did not provide any evidence to suggest that differences in the microbiota caused the variability.

Reviewer #3 (Remarks to the Author):

All in all this is a very nice study which is building on previous work in sensible but novel directions:

Recognizing microbial mediation as a possible mechanism of drug action in diabetes, the authors investigate this using a longitudinal setup, thus far rare but ideal for this investigation. They contrast metagenomic and molecular measurements with phenotypes and makes a strong case for acarbose effects on the gut microbiome acting via bile acid changes to improve health.

This research is timely, well-executed and relevant.

Necessary revision: I have some concerns which should be addressed before publication:

line 148-151: Is this test done for each of the 10M genes? If so, P-values must be FDR adjusted - are those shown here so adjusted?

line 171-183: Here, many features are described as significantly different in abundance. Again, P-values must then be FDR-adjusted for the number of tested features, were they?

line 211-212: Same here?

line 215: Should be Bacteroides?

line 242-262: It would be good with some more discussion on the role of the Prevotella and Bacteroidetes-Firmicutes dimensions here, and on whether these findings are expected.

line 393-396: I am concerned with the annotation of BA metabolism genes here. The criteria for E-value, identity and coverage are very lenient. There certainly are valid hits at those thresholds, but chances are many of these IGC genes are even more similar to other genes of similar folds which are not BA metabolism genes. This should be checked. Perhaps search the 30K hits against KEGG and verify the top hit is not annotated with something which is incompatible with involvement in BA metabolism? Or some similar approach. In any case, this part should be made more stringent.

Reviewers' comments:

Reviewer #1 (Remarks to the Author):

This paper by Gu et al describes the effect of acarbose vs glipizide on bile acid metabolism (asting plasma samples) between baseline and 3 months after start treatment in 106 naive Chinese DM2 patients (1:1 randomized, 94 of the enrolled patients completed the study 51 to acarbose and 43 to glipizide). Acarbose decreased plasma secondary bile acid and increased unconjugated primary bile acids.

Moreover, several correlations between bile acids and elated with improvements in HbA1c, BW, FLI, blood pressure and lipid profiles were observed.

Only acarbose treatment significantly affected fecal microbiota composition (predominantly saccharolytic *Lactobacillus* and *Bifidobacterium* species) that were also related to bile acid levels in plasma. Subsequent functional analyses showed that 7- α -hydroxysteroid dehydrogenase were significantly decreased in microbial genes after Acarbose, while the *baiG* (encoding bile acid transporter) and genes encoding 7- β -HSDH were enriched upon acarbose;

these were all related to *B. plebeius*, *B. vulgatus/dorei* and *B. dorei*. These findings were aligned by clustering of enterotypes with success of acarbose treatment (*bacteroides* cluster > *prevotella* cluster).

Overall this is an interesting study, however several major comments preclude publication of this paper and need to be addressed since they can induce major bias on the microbiota analyses and outcome.

1. Can the authors provide Dietary data during the study? This as diet is known to affect both microbiota composition (David, Nature 2014) and bile acids excretion (O'Keefe SJ, Nat Comm 2015 :6342)

We appreciated the reviewer's comment. In the study of O'Keefe SJ, et. al. ¹ the two studied

populations lived in very different environments (the United States vs Africa) and furthermore had extremely different dietary habits (low fibre vs high fibre intake). By contrast, the patients enrolled in this clinical trial were from different locations of Shanghai and most were long term residents with similar diet habits.

We have now included diet data recorded by food frequency query (FFQ) in the **Supplementary information** and **Supplementary data 19**. The FFQ is a Chinese translated version based on the SLAN study². Primarily, the distribution of vegetarian or non-vegetarian and dairy-food consumer or non-dairy-food consumer did not differ between the two arms at baseline (Pearson's Chi-squared test, $P>0.05$). The frequencies of vegetable, meat, fish, sweets intakes and the frequencies of eating out were similar in patients belonging to the two treatment arms pre- and post-treatment (Kruskal-Wallis rank sum test, $P>0.05$). More importantly, these dietary features of patients in both arms showed no significant changes after treatment (Kruskal-Wallis rank sum test, $P>0.05$). Hence, it is not likely that the observed changes in the microbiota after Acarbose treatment are mediated by changes in the diet. However, it should be noted that the self-reported FFQ has limitations³ and is not optimal to reflect the complicated composition of Chinese food or various ways of cooking.

2. Did the authors also determine changes in Fecal bile acids, if not can they provide these data. This as this would largely add to the understanding of which bacterial strains are involved in altered bile acids metabolism upon acarbose treatment.

We thank the reviewer for raising this point and agree that faecal BA composition might add to the interpretation/understanding. Accordingly, as suggested we determined the BA composition in the faecal samples in Acarbose arm. In brief, we found substantial lowered levels of SBAs in faecal samples after Acarbose treatment that aligned with the reduced abundances of key *bai* genes and plasma BA composition alterations, suggesting that Acarbose decreased gut microbial BA transformation. This is now included in the revised manuscript, page10, lines **221-231** “We investigated if the changes in the bacterial potential for BA metabolism induced by Acarbose treatment were reflected in the BA composition in

the faecal samples. The total level of faecal BAs was significantly decreased after Acarbose treatment. Actually, the faecal levels of the primary BAs, CA and CDCA, increased following Acarbose treatment, but this was more than counteracted by the large decrease in the levels of the secondary BAs, DCA and LCA. Overall, these changes resulted in an increased faecal PBA/SBA ratio in response to Acarbose treatment (**Supplemental Data 12, Supplemental Fig. 4**). The faecal UnconBA/conBA ratio was unchanged. These changes mirrored the changed relative abundance of *bai* genes with an unchanged relative abundance of *bsh* genes following Acarbose treatment (illustrated in **Fig. 4c**), but do not explain the significant increase in the plasma UnconBA/ConBA ratio after Acarbose treatment.”

We also added a comparison of faecal BAs between Cluster B and Cluster P, page 11, lines **249-253** “However, already at baseline patients in Cluster B had lower levels of LCA and DCA, and higher levels of ursodeoxycholic acid (UDCA) in the BA pool both in plasma and faeces(**Wilcoxon rank-sum test, P<0.05, Fig. 5a; Supplementary Data 15, 16**)”.

3. Since predominantly *B. plebeius*, *B. vulgatus/dorei* and *B. dorei*, this suggests that SCFA like butyrate are also involved Can the authors show SCFA levels in feces before and after treatment? can the authors provide these data to give more insight into mode of mechanism?

We thank the reviewer for this constructive suggestion. Accordingly, we determined the levels of butyrate in the faecal samples from patients treated with Acarbose. The determination of faecal butyrate content was performed using an Agilent 7890A gas chromatography system coupled to an Agilent 5975C inert MSD system (Agilent Technologies Inc., CA, USA). An OPTIMA® WAXplus fused-silica capillary column (30 m × 0.25 mm × 0.25µm; MACHEREY-NAGEL GmbH & Co.KG, Germany) was utilized to separate the derivatives. Helium (>99.999%) was used as a carrier gas at a constant flow rate of 1 mL/min through the column. This analysis showed that the median level of butyrate did not change significantly in response to treatment with Acarbose (median before treatment 43.3 nmol/g faeces, median post treatment 47.6 nmol/g faeces, p=0.096). We agree with the reviewer that *Bacteroides* species are important butyrate producers. However, the relative abundances of other abundant known butyrate producers, such as *Faecalibacterium prausnitzii*, and several species from

Roseburia, *Blautia* and *Eubacterium* did not change significantly after Acarbose treatment in our study (Supplementary Data 6). Further, both the KEGG pathway, map00650, involved in butanoate metabolism (Supplementary Data 13) and the key enzyme for butyrate production, K19709 (butyryl-CoA: acetate CoA transferase [EC:2.8.3.8]) did not change significantly after Acarbose treatment. Hence, these results might partly reflect a relatively stable butyrate production in human colon.

Additionally, as reported by previous studies, more than 95% of the produced butyrate are rapidly absorbed by the colonocytes, while the remaining 5% are secreted in the feces^{4,5}. Thus, the faecal SCFAs may not necessarily reflect butyrate levels in the more proximal colon and blood. Thus, further research is needed to establish the relation between butyrate levels blood and faecal samples and to determine clinical implications and interpretation of the content of faecal butyrate. Therefore, we decided not to include data on the faecal content of butyrate in the present manuscript.

4. Enterotypes quite controversial and debated in the field (Knights, Cell Host Microbe. 2014). It would be advisable to add such a reference this to the limitation section of this paper.

We are aware that the concept of enterotypes has been a matter of dispute. We have now added a sentence recognizing this, so lines **243-245** now reads: “Studies have suggested a stratification of individuals by clustering algorithms based on the genus composition of the intestinal microbiota^{36,37}, but the concept of the existence of discrete enterotypes has also been questioned suggesting a continuous distribution of enterotypes in an individual³⁸. In our study, the baseline microbiomes of patients could be clearly separated into two enterotype-like clusters....”. Thus, in the revised version, we use the wording enterotype-like clustering, which reflects the result of a bioinformatics analysis pointing to the existence of clusters here driven by the *Bacteroides* or *Prevotella* genus. We think that it would be beyond the scope of present manuscript to include a more lengthy discussion of the enterotype concept, and we hope that the addition is palatable to the reviewer.

Reviewer #2 (Remarks to the Author):

Gu et al. examined the response of gut microbiota and plasma BAs to the anti-diabetic drugs Acarbose and Glipizide in naïve type 2 diabetic (T2D) patients. The authors report that Acarbose specifically, and not Glipizide, increased unconjugated BAs and the primary/secondary BA ratio, increased Lactobacillus and Bifidobacterium abundance, whilst decreasing Bacteroides abundance. The authors observed that these changes were dependent on the host microbiome at baseline prior to treatment. Patients with baseline Bacteroides enterotype exhibited an improved response to Acarbose than those with Prevotella. These findings suggest that baseline microbiota enterotype may be used as a stratification for anti-diabetic treatments.

Whilst the suggestion of stratification of patients based on enterotype to predict response to anti-diabetic treatment is an intriguing possibility, there are some concerns with the statistical data analysis, cluster analysis and the presentation of the data.

Comments:

1. Patients recruited from 5 centers. Was this taken into account when measuring the baseline enterotypes. PCA analysis of enterotype of all patients to ensure no segregation based on center recruitment.

We understand the Reviewer's concern. We compared the distribution of microbiome clusters across the centres. As shown by a PCA at genus level, we did not observe differences in clustering of the microbiomes between the five centres.

A chi-squared test based on the number of patients from 5 centres in two clusters showed no significant segregation in relation to centre recruitment ($P=0.287$).

2. Standard units for lipid measurements are mg/dL. Graphs in Figure 1 should be changed to reflect these units. The normal range for healthy and naïve T2D should also be indicated.

In all five hospitals, we are as recommended using SI unit, i.e. mmol/L for lipid measurement in routine biochemical test. If Nature Communication wants to keep the non-SI units, we will of course convert the units of lipids following the reviewers' suggestion (TG mmol/L \times 88.5=1 mg/dL, TC 1 mg/dL = 38.61 mmol/L). However, honestly, we prefer SI units.

3. The presentation of the data in **Figure 1d** is extremely confusing. It would be more

appropriate to have separate graphs for Acarbose and Glipizide.

To clarify we have revised Figure 1d and 1e and have separated the Acarbose and Glipizide group in the graphs.

4. There is no label on the y-axis of the HOMA-IR graph in Figure 1f.

Thanks for this notion. We have added “uIU*mmol” as the unit of the HOMA-IR graph in Figure 1f.

5. The data of Table 1 show that the effects of Acarbose are greater than the effects of Glipizide. However, the magnitude of this effect is the same in baseline measurements, which are stated as being not significant. Additionally, a difference of 2kg with an SEM of 10 seems unlikely to be significant. Are the data really non-parametric? More information about the statistical analysis is needed, and preferably plots of the individual data points should be included.

We thank the reviewer for this constructive comment. As shown in **revised Supplementary Data 2**, Shapiro–Wilk tests revealed that most of clinical measurements showed non-normal distributions. We have also amended the Post-Treat P values in **Table 1**.

We apologize for the confusion concerning the p value in the table. The p values listed in **Table 1** in the column representing post-treatment refers to the changes in clinical parameters after treatment taken from the Supplemental Data 2 and not to post treatment differences between the 2 arms. We have now included the p values for comparing post treatment values in the **Table 1** and added them in **Supplemental Data 2** in the red colour. As indicated, there were no significant differences in body weight between patients after receiving Acarbose and Glipizide.

6. The BA data in Table 2 is extremely confusing. **Considering the total BA concentrations do not change, but the BA pool composition is significantly altered, a more appropriate way to display this data is either in box plot or pie chart.**

We thank the reviewer for this suggestion. **Table 2** is now the **Supplemental data 3**. We plotted the absolute values of BAs in **Figure 2a** as a box plot, and BA species composition in **Figure 2b** as a bar plot.

7. There is no label on the y-axis of Figure 3a. Is this gene number? Or abundance? The reference in the text would indicate gene number.

We apologized for the confusion. The y-axis of **Figure 3a** indicates gene number. We have now labelled the y-axis

8. What are the different coloured bars in Figure 3e? There is no reference in the text, and no legend on the graph.

We apologize for the confusion.

The legend to Figure 3e has now been revised including a colour code to better convey the message (Page 35 Lines **708-712**).

9. What do the two colors of mOTU indicate? Are they ranked in some way? The boxes to the left indicate the bacterial phyla, but it is not clear what this other separate segregation is.

We apologize for the confusion.

We ordered the mOTUs according to their relative abundances at the baseline. mOTUs in red colour represent mOTU that decreased in abundance after Acarbose treatment, whereas mOTUs in green colour represent mOTUs that increased in abundances after Acarbose treatment. The mOTUs we present in **Figure 3** are those that changed significantly in abundance after treatments ($q < 0.01$). The legend to Figure 3 has been revised accordingly.

10. In the cluster analysis, was the distribution of cluster B and cluster P equally distributed throughout the 5 centres? Was this taken into account during the cluster analysis?

Yes, as mentioned in the response to question 1, we performed a Chi-square test to show that there were no significant differences in microbiome clustering between the five centres.

11. Some references are not correct. For example, references **4, 5 and 8** are used as citations for the effect of the microbiome post gastric bypass surgery. However, the data in these citations states that reductions in diabetes post gastric bypass could not always be explained by weight loss but did not provide any evidence to suggest that differences in the microbiota caused the variability.

We apologize for the errors. We have deleted references 4-6:

4. Sjostrom L, et al. Lifestyle, diabetes, and cardiovascular risk factors 10 years after bariatric surgery. *The New England journal of medicine* 351, 2683-2693 (2004).
5. Schauer PR, et al. Bariatric surgery versus intensive medical therapy for diabetes--3-year outcomes. *The New England journal of medicine* 370, 2002-2013 (2014).
6. Mingrone G, et al. Bariatric surgery versus conventional medical therapy for type 2 diabetes. *The New England journal of medicine* 366, 1577-1585 (2012).

The original references 8-9 (see below) are now references 4-5

8→4. Aron-Wisnewsky J, Dore J, Clement K. The importance of the gut microbiota after bariatric surgery. *Nature reviews Gastroenterology & hepatology* 9, 590-598 (2012).

9→5. Liou AP, Paziuk M, Luevano JM, Jr., Machineni S, Turnbaugh PJ, Kaplan LM. Conserved shifts in the gut microbiota due to gastric bypass reduce host weight and adiposity. *Science translational medicine* 5, 178ra141 (2013).

Finally, we added two recent publications on the link between metformin treatment and microbiome alteration as the new references 6-7,

6. Wu H, et al. Metformin alters the gut microbiome of individuals with treatment-naive type 2 diabetes, contributing to the therapeutic effects of the drug. *Nature medicine*, (2017).

7. Forslund K, et al. Disentangling type 2 diabetes and metformin treatment signatures in the human gut microbiota. *Nature*, (2015).

Reviewer #3 (Remarks to the Author):

All in all this is a very nice study which is building on previous work in sensible but novel directions: Recognizing microbial mediation as a possible mechanism of drug action in diabetes, the authors investigate this using a longitudinal setup, thus far rare but ideal for this investigation. They contrast metagenomic and molecular measurements with phenotypes and makes a strong case for acarbose effects on the gut microbiome acting via bile acid changes to improve health. This research is **timely, well-executed and relevant.**

We thank the reviewer for the positive comment.

Necessary revision: I have some concerns which should be addressed before publication:

line 148-151: Is this test done for each of the 10M genes? If so, P-values must be FDR adjusted - are those shown here so adjusted?

We thank the reviewer for this piece of advice.

We have revised the sentence as “Metagenomic analyses demonstrated that Acarbose treatment significantly altered the relative abundances of 141,382 microbial genes, in contrast, no genes were altered by Glipizide treatment (paired Wilcoxon rank-sum test, $q < 0.01$) (Supplementary Data 5)” (Lines **148-150**).

line 171-183: Here, many features are described as significantly different in abundance. Again, P-values must then be FDR-adjusted for the number of tested features, were they?

Again, we have revised the manuscript by using q value (FDR-adjusted P-values) to determine the significance

Lines 173-177, “In the untreated T2D patients at the baseline permutation multivariate

analysis of variance (PERMANOVA) analysis revealed that plasma levels of the SBAs, ursodeoxycholic acid (UDCA), glyoursodeoxycholic acid (GUDCA), lithocholic acid (LCA) and deoxycholic acid (DCA) correlated with differences in the composition of the gut microbiota (Bray-Curtis distance, $q < 0.01$; **Supplementary Data 7**).”

Lines **177-181**, less mOTUs correlated with BA changes following Acarbose treatment, when $q < 0.05$. Thus, we changed our text “Interestingly, 58 out of the 69 mOTUs that exhibited significant changes in relative abundance in response to Acarbose treatment also correlated with Acarbose-induced alterations in plasma BA composition (GEE corrected for age, sex and BMI, $q < 0.05$; **Supplementary Data 8**), pointing to a tight link between Acarbose-dependent alterations of the gut microbiota and plasma BA composition.”

line 211-212: Same here?

Lines **208-209**, the sentence now reads “None of these genes changed significantly in relative abundance after Glipizide treatment (paired Wilcoxon rank-sum test, $P > 0.05$; **Fig.4a**; **Supplementary Data 10**)”

line 215: Should be *Bacteroides*?

We apologize for the error. The text has been change to *Bacteroides*.

line 242-262: It would be good with some more discussion on the role of the Prevotella and Bacteroidetes-Firmicutes dimensions here, and on whether these findings are expected.

We thank the reviewer for this suggestion, but we are a little uncertain as to what the reviewer wants to discuss. As shown in **Supplemental Figure 5** and **Supplemental Figure 6**, the species of the predominant genera in Cluster B were more susceptible to change in abundance by the Acarbose treatment, including *Bacteroides* and *Bifidobacterium*. The species belonging to these two genera are major BA gene contributors pre- and post-treatment, respectively, as

we show in **Supplemental Data 11** and **Figure 4**. Long-term intake of an animal protein-based diet (rich in protein and fat) has been suggested to correlate with *Bacteroides spp.* and other species related to it⁶ and plant-based diet (rich in fibre) correlate with *Prevotella*. An animal protein-based diet and a dietary change from plant-based to animal protein-based diet can increase BA deconjugation and secondary BA formation and thereby increase the level of secondary BAs¹. Accordingly, gut microbiome of patients of Cluster B contained more BA metabolizing species, and could thereby elicit changes in the pool of BAs modulating BA signalling that may mediate the additional metabolic benefits of Acarbose. The results of the faecal BA analysis further supported the different BA metabolizing potential of the different microbiome clusters.

line 393-396: I am concerned with the annotation of BA metabolism genes here. **The criteria for E-value, identity and coverage are very lenient.** There certainly are valid hits at those thresholds, but chances are many of these IGC genes are even more similar to other genes of similar folds which are not BA metabolism genes. This should be checked. Perhaps search the **30K hits against KEGG and verify the top hit is not annotated with something which is incompatible with involvement in BA metabolism?** Or some similar approach. In any case, **this part should be made more stringent.**

We thank the reviewer for the expert comments on annotation of genes involved in BA metabolism.

Firstly, since the reference protein sequence annotated as chenodeoxycholytaurine hydrolase (EC 3.5.1.74) recently was removed from the UniProt database, we omitted the results for genes potentially encoding EC 3.5.1.74.

Further, we have improved our BlastP cutoffs now using an E value $< 1E-5$ and a bit score > 60 , resulting in 16,102 annotated genes. These improved annotation parameters are more stringent than those used in a previous study dealing with functional annotation of genes associated with bile acid metabolism⁷. Additionally, we have assessed the 16,102 potential hits against KEGG using the KEGG online BlastKOALA algorithm⁸.

As shown in Supplemental Data 20, the filtered genes annotated to *bsh*, *baiE* and *baiI* by our pipeline showed high consistency with the KEGG online results, with 69%-85% genes assigned to the same KO. For genes annotated to *baiG* (Bile acid transporter), 72.2% were assigned to K03453 (bile acid: Na⁺ symporter, BASS family), although no KO was reported and annotated to *baiG*. For genes annotated to *7β-hsdh* by our pipeline, 92.3% were assigned to K07124.

By contrast, the remaining Uniprot database annotated genes showed rather low consistency with the KEGG online database. For example, genes annotated to *baiF* (K15871, EC 3.1.2.26, Bile acid-CoA hydrolase) by Uniprot were largely assigned to K18702 (uctC; CoA:oxalate CoA-transferase [EC:2.8.3.19], 43.4%), K07749 (frc; formyl-CoA transferase [EC:2.8.3.16], 31.0%) and K08298 (caiB; L-carnitine CoA-transferase [EC:2.8.3.21], 13.3%) by KEGG BlastKOALA. For genes annotated to *baiA* (K15869, EC 1.1.1.395, 3 α -hydroxy bile acid-CoA-ester 3-dehydrogenase) by Uniprot, more than 80% were assigned to K00059 (fabG; 3-oxoacyl-[acyl-carrier protein] reductase [EC:1.1.1.100]) by KEGG.

There are several possibilities explaining the inconsistency between two methods.

First, the high sequence identity between target KOs (enzymes) and other KOs (enzymes). For example, the KEGG annotated CoA-transferases exhibited very high identity with Bile acid-CoA hydrolase. The protein sequence A0A0S2W5Y8 (EC2.8.3.16, K07749) exhibited 99.7% identity with sequence A0A1C6GV35 (EC 3.1.2.26, Bile acid-CoA hydrolase); the sequence F7UY98 (K08298) exhibited 92.3% identity with sequence T4BVW9 (EC 3.1.2.26, Bile acid-CoA hydrolase).

Second, there are differences between the two annotation algorithms. For example, the gene encoding protein sequence 1 (ID: V1.CD3-0-PN_GL012122) was annotated to *baiA* ((K15869) by Uniprot but assigned to K00059 by KEGG, although protein sequence 1 exhibited higher identity (87%) with ag:ACF20977 (K15869) than with sequences from K00059(**Supplemental Data 21**).

Finally, the inconsistency could be also caused by unique sequences in the two databases. For instance, 12 of 13 genes annotated to *7β-hsdh* (7-beta-hydroxysteroid dehydrogenase, EC 1.1.1.201) were assigned to K07124 by KEGG. The protein sequences encoded by these genes exhibited more than 70% identity with R9UAM1 (7-beta-hydroxysteroid

dehydrogenase, EC 1.1.1.201), and less than 40% identity with sequences from K07124(Supplemental Data 22).

Again, we thank the reviewer for raising this very important point on functional annotation in metagenomics study. Although bit score cutoffs (Score > 60)^{9,10} and E value cutoff⁷ (E value > 1E-5) have been widely used in previously published studies, our analyses clearly reveal the limitations to distinguish different enzymes/functions with high sequence identity. Thus, potentially annotated genes in our results should be further validated using sequencing of bacterial isolates. Accordingly, in the present study only genes exhibiting highly consistent annotation (*bsh*, *baiE*, *baiG*, *baiI* and *7β-hsdh*) have been included in the analyses.

We have updated data on gene abundancies and species contributor calculation in Supplemental Data 10,11,17, Figure 4, and added Supplemental Data 20, 21 and 22.

1. O’Keefe, S. J. D. *et al.* Fat, fibre and cancer risk in African Americans and rural Africans. *Nat. Commun.* **6**, 6342 (2015).
2. Harrington, J. *et al.* Sociodemographic, health and lifestyle predictors of poor diets. *Public Health Nutr.* **14**, 2166–75 (2011).
3. Hodson, L., Skeaff, C. M. & Fielding, B. A. Fatty acid composition of adipose tissue and blood in humans and its use as a biomarker of dietary intake. *Progress in Lipid Research* **47**, 348–380 (2008).
4. Ruppin, H., Bar-Meir, S., Soergel, K. H., Wood, C. M. & Schmitt, M. G. Absorption of short-chain fatty acids by the colon. *Gastroenterology* **78**, 1500–7 (1980).
5. Topping, D. L. & Clifton, P. M. Short-chain fatty acids and human colonic function: roles of resistant starch and nonstarch polysaccharides. *Physiol. Rev.* **81**, 1031–1064 (2001).
6. Wu, G. D. *et al.* Linking long-term dietary patterns with gut microbial enterotypes. *Science* **334**, 105–8 (2011).
7. Tremaroli, V. *et al.* Roux-en-Y Gastric Bypass and Vertical Banded Gastroplasty Induce Long-Term Changes on the Human Gut Microbiome Contributing to Fat Mass

- Regulation. *Cell Metab.* **22**, 228–238 (2015).
8. Kanehisa, M., Sato, Y. & Morishima, K. BlastKOALA and GhostKOALA: KEGG Tools for Functional Characterization of Genome and Metagenome Sequences. *Journal of Molecular Biology* **428**, 726–731 (2016).
 9. Tringe, S. G. Comparative Metagenomics of Microbial Communities. *Science (80-)*. **308**, 554–557 (2005).
 10. Harrington, E. D. *et al.* Quantitative assessment of protein function prediction from metagenomics shotgun sequences. *Proc Natl Acad Sci U S A* **104**, 13913–13918 (2007).

Reviewers' comments:

Reviewer #1 (Remarks to the Author):

the authors have answered my questions in a satisfactorily manner

Reviewer #2 (Remarks to the Author):

To the Authors:

Gu et al. have adequately addressed all of this reviewers concerns.

Reviewer #3 (Remarks to the Author):

My concerns are largely addressed, except one, which may or may not have been dealt with also.

Specifically, the assignment of BA genes. The authors show that testing against KEGG produces sometimes the same, sometimes different results. This is fine. The changed cutoffs might still not necessarily say much (because as the authors show some of these families have different function at low sequence divergence). However, there is only so much any one study can do about this.

What I think then is necessary (and where I don't know yet whether the authors did this) is that when annotating IGC genes against UniProt to the BA functions, not only is the threshold for similarity achieved, but there is no better-scoring hit to anything in UniProt which has another, non-BA functional annotations. That is to say, if this search was done comparatively, against the whole of the database, and the best hit taken, then my concern is fully alleviated. Whereas if the search was done only against the BA exemplars (so the risk of something else with different annotation scoring higher remains), then this issue remains to be dealt with. Could you clarify whether this mapping was competitive (i.e. search done against not only BA genes but other genes too, so that best hit to a BA gene precludes a better hit to something else) or not? If that holds, then never mind discrepancy between UniProt and KEGG.

Reviewer #3 (Remarks to the Author):

My concerns are largely addressed, except one, which may or may not have been dealt with also.

Specifically, the assignment of BA genes. The authors show that testing against KEGG produces sometimes the same, sometimes different results. This is fine. The changed cutoffs might still not necessarily say much (because as the authors show some of these families have different function at low sequence divergence). However, there is only so much any one study can do about this.

What I think then is necessary (and where I don't know yet whether the authors did this) is that when annotating IGC genes against UniProt to the BA functions, not only is the threshold for similarity achieved, but there is no better-scoring hit to anything in UniProt which has another, non-BA functional annotations. That is to say, if this search was done comparatively, against the whole of the database, and the best hit taken, then my concern is fully alleviated. Whereas if the search was done only against the BA exemplars (so the risk of something else with different annotation scoring higher remains), then this issue remains to be dealt with. Could you clarify whether this mapping was competitive (i.e. search done against not only BA genes but other genes too, so that best hit to a BA gene precludes a better hit to something else) or not? If that holds, then never mind discrepancy between UniProt and KEGG.

We understand the reviewer's concern and thank the reviewer for the constructive suggestion.

In our previously submitted revised version, we applied a computation-efficient strategy to identify genes encoding putative BA metabolism functions and confirmed our results by searching these sequences against the KEGG database. We are happy to confirm these findings again using the complete UniProt Database upon the request of the reviewer.

According to reviewer's suggestion, we further searched the 16K genes against the entire UniProt database (Release 2017_07 of 05-Jul-2017) and calculated the percentage of genes with the best-scoring hits assigned to the same BA metabolism functions/enzymes identified in the initial search against the 1,011 amino acid sequences annotated as enzymes involved in BA metabolism in the UniProt database (Supplementary Data 23).

It should be emphasized that the names for genes/proteins are highly ambiguous in UniProt database and there are usually multiple names for the same gene or protein. This is also noted in the UniProt database stating that the accuracy of the "submitted names" relies on the information provided by the submitter of the nucleotide entry.

http://www.uniprot.org/help/different_protein_gene_names). Furthermore, many proteins are submitted as uncharacterized proteins. Thus, it is impossible to arrive at completely unified annotations of the UniProt best hits.

However, for *bsh*, *baiE*, *baiG*, *baiI* and *7 β -hsdh*, the annotation of the highest scoring UniProt hits representing more than 85% of the genes was consistent with our previously submitted version, after removing the uncharacterized proteins and combining the different synonymous protein names.

For *baiF*, *baiCD*, *baiH*, *baiB*, *baiA* and *7 α -hsdh*, their highest scoring UniProt hits were largely assigned to non-BA metabolism functions.

In conclusion, we agree with the reviewer that the improved BlastP cutoffs in our study might still not suffice, and thus, does not assure prediction accuracy of different protein family members with high sequence similarity. Still, the robustness of the annotation of *bsh*, *baiE*, *baiG*, *baiI* and *7 β -hsdh* genes, emphasized in the manuscript, strongly suggests correct functional annotation of these genes. We hope our response has satisfactorily addressed the reviewer's concern. We have revised the Supplementary Information accordingly (line 73-87) and further added Supplementary Data 23 (line 196-197) in the Supplementary Dataset list.

REVIEWERS' COMMENTS:

Reviewer #3 (Remarks to the Author):

My concerns have now been addressed.

REVIEWERS' COMMENTS:

Reviewer #3 (Remarks to the Author):

My concerns have now been addressed.

Answer: We thank the reviewer.